# MMGeoLM: Hard Negative Contrastive Learning for Fine-Grained Geometric Understanding in Large Multimodal Models

## Abstract

Large Multimodal Models (LMMs) typically build on ViTs (e.g., CLIP), yet their training with simple random in-batch negatives limits the ability to capture fine-grained visual differences, particularly in geometric scenarios. To address this challenge, we propose a novel **hard negative contrastive learning** framework for the vision encoder, which combines *image-based contrastive learning* using generation-based hard negatives created by perturbing diagram generation code, and *text-based contrastive learning* using rule-based negatives derived from modified geometric descriptions and retrieval-based negatives selected based on caption similarity. We train a vision encoder (CLIP) using our hard negative training method, namely MMCLIP (Multimodal Math CLIP), and subsequently train an LMM for geometric problem-solving. Experiments show that our trained model, MMGeoLM, significantly outperforms other open-source models on three geometric reasoning benchmarks. Even with a size of 7B, it can rival powerful closed-source models like GPT-4o. We further conduct ablation studies to analyze three key factors: hard negative types, the efficiency of image-based negatives, and training configurations. These analyses yield important insights into optimizing the training pipeline of vision encoder for fine-grained geometric reasoning tasks.

## 1 Introduction

Geometric mathematical reasoning has garnered significant attention as an essential capability for large multimodal models (Anthropic, 2024; OpenAI, 2023; Bai et al., 2023). It requires fine-grained identification of visual elements (Lu et al., 2023) within the given images, such as geometric shapes, spatial configurations, and the relationships between them (He et al., 2024).

However, the "eyes" of most existing LMMs, i.e., their pretrained vision encoders such as CLIP (Patel et al., 2024; Yang et al., 2023; Goel et al., 2022), are primarily trained on general visual datasets that do not emphasize the intricate features necessary for specialized mathematical reasoning. Therefore, these models often fail to understand the nuanced geometric information accurately and produce incorrect reasoning and answers. As shown in Figure 1, facing a simple parallel line problem, the leading LMMs such as GPT-4o (OpenAI, 2024a), Claude-3 (Anthropic, 2024), and Qwen2.5-VL (Bai et al., 2025) all hallucinate non-existent elements or misinterpret spatial relationships (e.g., $\triangle ABC$, $\triangle ABE$, and the concept of similar triangles), exhibiting notable deficiencies in capturing the intricate geometric details.

To address these shortcomings, recent research has focused on strategies such as fine-tuning on specialized mathematical datasets (Gao et al., 2023; Zhang et al., 2024c; Peng et al., 2024a;b) or utilizing massive image-caption pairs to enhance the visual perception ability of LMMs (Qi et al., 2024; Wang et al., 2024) by aligning images with corresponding captions. However, relying solely on positive image-caption pairs may lead to spurious alignment, with models may behave like "bag of words" rather than achieving semantic understanding (Zhang et al., 2024b; Doveh et al., 2023a). To achieve more robust and semantically meaningful alignment, it is crucial to incorporate hard negative samples—semantically similar but mismatched pairs—which push the vision encoder to learn finer distinctions beyond shallow correlations. Therefore, to further improve LMMs' abilities

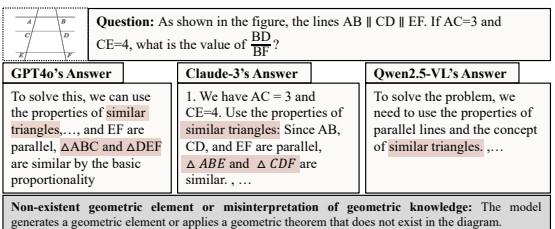

| Model | Training | Training Data | MM-Math |
|-------|----------|---------------|---------|
| AltCLIP | No | No | 23.8 |
| AltCLIP | In-Batch | Randomly-sampled Neg (400K) | 24.6 |
| AltCLIP | MMCLIP | ①: Retrieval Neg (100K) | 26.6 |
| AltCLIP | MMCLIP | ②: Rule Neg (100K) | 28.1 |
| AltCLIP | MMCLIP | ③: Image Neg (4K) | 28.2 |
| AltCLIP | MMCLIP | ①+② | 28.4 |
| AltCLIP | MMCLIP | ①+②+③ | **30.1** |

Figure 1: Examples of hallucination: non-existent elements and relation misinterpretation.

Table 1: Comparison of geometric reasoning performance on MM-Math across LMMs with vision encoders trained with different hard negative data.

for capturing geometric information, we investigate a key question: **How can we systematically construct hard negatives tailored for geometric reasoning?**

In this work, we propose two types of hard negative sample construction methods, i.e., *image-based* and *text-based*, to enhance fine-grained geometric element recognition in vision encoder. For text-based contrastive learning, we design two strategies to create negative captions for a geometry image: (1) a *retrieval-based* method that employs dense retrieval in a geometric-domain text corpus to select captions with high lexical similarity but differing content as negative samples; and (2) a *rule-based* method that modifies key geometric attributes in the captions, such as shapes, angles, and lengths, while keeping other elements unchanged, thereby producing negative samples that bear similar appearance but with key distinct information from the positives. For image-based contrastive learning, we introduce a novel method that leverages a large language model (LLM) to generate a detailed caption and corresponding diagram generation code for a given geometry problem, forming the positive image sample. The LLM then perturbs the code to create visually similar but geometrically incorrect diagrams, which serve as hard negative samples. Additionally, we modify the original CLIP training framework and propose MMCLIP, a method designed to handle an arbitrary number of hard negative samples centered around a single image or caption. Table 1 shows the effectiveness of MMCLIP training on different hard negative sets.

We evaluate our trained MMGeoLM on four geometric benchmarks spanning two categories: choice-based questions (GeoQA, MathVista, and We-Math) and open-ended questions (MM-Math). Results show that the proposed model outperforms all existing open-source models on GeoQA and MathVista, and achieves state-of-the-art performance on MM-Math, surpassing GPT-4o by 7.4%. Ablation studies confirm both text-based and image-based hard negatives benefit model performance. Notably, using only 4K image-based negatives yields better results than 100K retrieval-based ones. These findings demonstrate that our hard negative construction and training strategy significantly enhances reasoning accuracy in geometric reasoning.

## 2 HARD NEGATIVE DATA CONSTRUCTION

### 2.1 NEGATIVE IMAGES CONSTRUCTION

We construct negative geometric examples by generating geometric diagram code for positive images and perturbing it to create negatives. Existing generation methods (Zhang et al., 2024c; Zou et al., 2024; Wei et al., 2024) rely on handcrafted rules that cover limited elements and overlook inter-element relationships, resulting in a gap from real-world problems. In contrast, authentic geometry problems (Sun et al., 2024) are more diverse and accurate, with geometric elements described either explicitly (e.g., "AB is 8 cm") or implicitly through reasoning (e.g., using the Pythagorean theorem), and reflected in the corresponding geometric diagrams (McClintock et al., 2002). Benefiting from advancements in mathematical reasoning and code-generation capabilities of LLMs (OpenAI, 2024b), which can effectively parse geometric conditions, perform necessary reasoning, and generate executable code to produce geometric diagrams closely matching the original images. As illustrated in Figure 2, providing the LLM with both the problem and the answer yields code and a caption. Executing the code produces diagrams that not only replicate the original figures but may also include additional *numeric markings*.

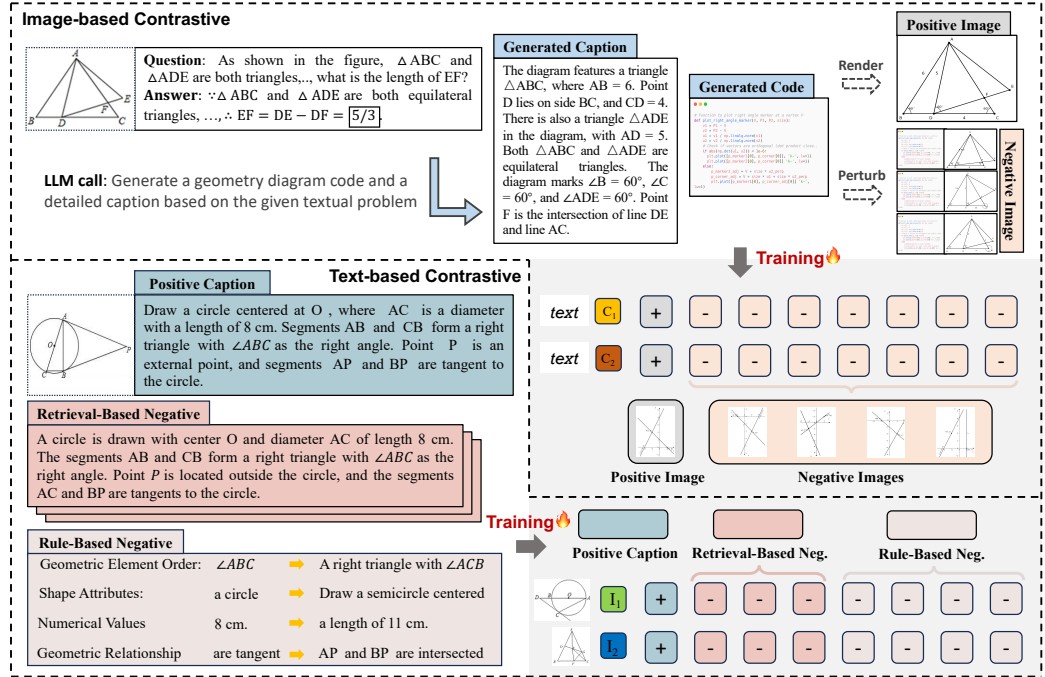

Figure 2: Image-based and text-based hard negative construction and the corresponding MMCLIP training method.

To handle occasional errors in model-generated code, we adopt a lightweight model (GLM et al., 2024) to correct syntax issues when the code fails. Subsequently, we generate detailed captions directly from code using Gemini 2.5 Pro (DeepMind, 2025). Upon evaluating 123 generated captions, three independent annotators confirmed that the accuracy of captions in representing geometric elements reaches 100%. Further details are provided in Appendix F.

To create the negative images, we design prompts for LLMs to generate 10 negative captions for each positive caption, differing subtly but critically. We then employ Gemini 2.5 Pro to modify the Python scripts based on these perturbed captions. These modified scripts produce diagrams that closely resemble the originals while aligning with the intended constraints. Representative examples of the constructed image-based negatives are provided in Appendix B.

## 2.2 NEGATIVE CAPTIONS CONSTRUCTION

We propose two methods for constructing text-based hard negative samples. The first method retrieves captions similar to each positive example from a large image-caption dataset. The second method generates targeted hard negative samples based on reasoning errors observed during LMMs' inference.

**Retrieval-based Hard Negative.** Many previous works in open-domain QA take the top-ranked instances recalled by the retriever as negative examples to further improve retriever performance (Karpukhin et al., 2020; Xiong et al., 2020; Huang et al., 2020). However, they suffer from the risks of introducing false negatives, which means some related instances are incorrectly treated as negatives (Xiong et al., 2020; Zhou et al., 2022; Yang et al., 2024). To address this issue, we build upon the recently proposed Mavis dataset (Zhang et al., 2024c), which mitigates false negatives by ensuring a one-to-one correspondence between each image and its caption. We use SimANS (Zhou et al., 2022) to encode all captions, compute pairwise similarities, and retrieve the top 100 most similar captions per image as hard negatives.

**Rule-based Hard Negative.** Existing studies modify positive captions by randomly altering nouns, attributes, and relationships (Yuksekgonul et al., 2022; Zhang et al., 2024a), while others

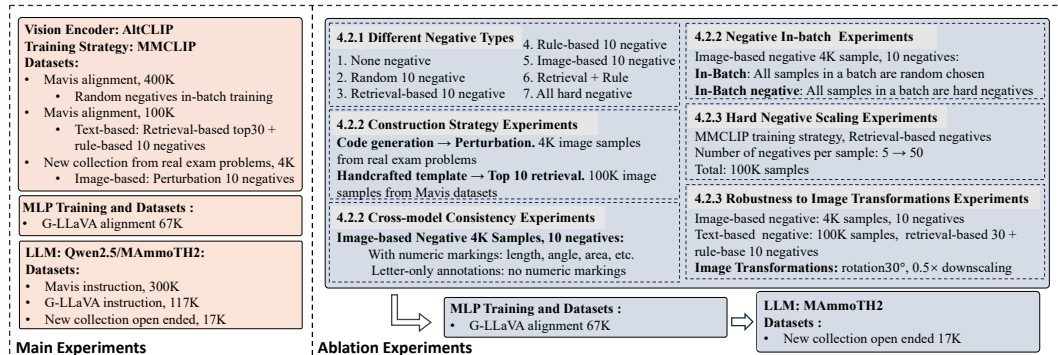

Figure 3: Overview of the MMGeoLM training pipeline, including main and each ablation experiment configurations, training strategies, and datasets.

leverage in-context learning to generate meaningful modifications (Patel et al., 2025). However, these methods either introduce semantic ambiguity or focus solely on the vision encoder level, without considering downstream reasoning errors, as exemplified by the misunderstanding in Figure 1, where parallel lines are mistaken as forming a triangle.

To address this issue, we analyze the evaluation results of LMMs on MM-Math (Sun et al., 2024) and identify four major types of image element recognition errors:

1. *Geometric element ordering*: Modifying the sequence of alphabetical order in geometric diagrams while ensuring the new order does not match the original cyclically (e.g., $ABCD$ changing to $CDAB$ is invalid).
2. *Shape attributes*: Altering properties such as changing squares to rectangles or right triangles to isosceles triangles.
3. *Geometric relationships*: Modifying relationships such as parallelism between two lines or similarity between two triangles.
4. *Numerical values*: Adjusting numerical values in captions, such as modifying angles or segment lengths.

To address these error types, we design rule-based strategies that use GLM-4 (GLM et al., 2024) to modify each positive caption and generate 10 hard negative examples. These text-based negatives are used in model training, and two representative types are shown in Figure 2.

## 3 MMGeoLM: Architecture and Training

### 3.1 Architecture

We adopt the LLaVA architecture (Liu et al., 2024), which comprises three components: (1) a vision encoder, (2) a 2-layer MLP adapter, and (3) an LLM backbone. For the LLM backbone, we use MAmmoTH2-7B (Yue et al., 2024) and Qwen2.5-7B-Instruct (Qwen et al., 2025). The vision encoder is based on AltCLIP (Chen et al., 2022b), configured with a maximum length of 512 tokens and a model size of 0.5B parameters.

### 3.2 Training Pipeline

Our training pipeline, illustrated in Figure 3, consists of three stages. **(1)** We first train CLIP with our MMCLIP training strategy detailed in Section 3.3. For text-based hard negatives, we adopt the Mavis image-caption alignment dataset (Zhang et al., 2024c), while for image-based negatives, we collect 4K geometry questions from real middle-school exams. Additional details, including the negative types and negative-to-positive ratio, are provided in Figure 3, and further statistics across categories of hard negatives are reported in Appendix A. **(2)** In the second stage, we fine-tune the MLP adapter using 67K image-text pairs from the G-LLaVA dataset (Gao et al., 2023). **(3)** In the third stage, we conduct supervised fine-tuning (SFT). We construct a high-quality training

dataset of 17K geometry problems from the `21st Century Education`[1]. This dataset aligns with middle school curricula and reflects real-world student assessment scenarios. Each problem includes a detailed analysis, from which we extract the core solution to create a structured step-by-step reasoning process. The final answers are enclosed in `\boxed{}`, enabling targeted learning. For our main experiments, we use a combined SFT dataset comprising 300K Mavis instruction data (Zhang et al., 2024c), 117K G-LLaVA instruction data (Gao et al., 2023), and the 17K collected open-ended geometry questions. For ablation studies, we use only the 17K geometry problems for efficient validation. We refer to the final trained model as `MMGeoLM`.

### 3.3 HARD NEGATIVE CLIP TRAINING

**In-batch Training**. Conventional CLIP-trained vision encoders (Radford et al., 2021; Doveh et al., 2023b) typically combine random examples for a batch and adopt *in-batch training loss over all the samples* in each batch. For each batch containing samples $\{(I_i, T_i)_{i=1}^N\}$, where each $I_i, T_i$ represents a pair of image and text caption, the loss is calculated as

$$\mathcal{L} = \frac{1}{N} \sum_{i=1}^{N} \left[ -\ln \frac{\exp(s(I_i, T_i))}{\sum_j \exp(s(I_i, T_j))} - \ln \frac{\exp(s(I_i, T_i))}{\sum_j \exp(s(I_j, T_i))} \right], \tag{1}$$

where $s(\cdot, \cdot)$ denotes the similarity function over the feature encodings.

**MMCLIP Training**. In our data construction process, hard negative captions and images are generated for each image-caption pair without providing corresponding positive samples for these hard negatives. As a result, our MMCLIP training derives the *loss solely from one positive sample and its associated hard negatives* within each batch. This guides the model to discern subtle differences, enhancing its fine-grained geometric understanding. Formally, given a text-based negative training batch $\{I, T^+, T_i^- |_{i=1}^N\}$, where $I$ represents the image, $T^+$ denotes the positive caption and $T_i^-$ is the negative caption conducted around the image. The loss function is formulated as follows:

$$\mathcal{L} = -\ln \frac{\exp(s(I, T^+))}{\exp(s(I, T^+)) + \sum_{i=1}^{N} \exp(s(I, T_i^-))}. \tag{2}$$

Since negative samples are constructed around elements within the image, their score $s(I, T_i^-)$ is expected to be higher than a randomly chosen one. From the above equation, we calculate the gradient for positive and negative samples:

$$\frac{\partial \mathcal{L}}{\partial s^+} = -\frac{\sum_{i=1}^{N} \exp(s_i^-)}{\exp(s^+) + \sum_{i=1}^{N} \exp(s_i^-)} \qquad \frac{\partial \mathcal{L}}{\partial s_i^-} = \frac{\exp(s_i^-)}{\exp(s^+) + \sum_{i=1}^{N} \exp(s_i^-)}$$

where $s^+ = s(I, T^+)$ and $s_i^- = s(I, T_i^-)$. Since the number of hard negatives is unrestricted, the gradient functions can be effectively optimized, leading to improved image-caption alignment. The training strategy is illustrated in Figure 2, where we also provide image-based negatives training batches $\{T, I^+, I_i^- |_{i=1}^M\}$, the loss can be derived similarly to Equation 2. Unlike in-batch training, which relies on random batch-internal negatives, MMCLIP allows training with independently constructed negatives, making it suitable for unimodal negatives, such as rule-based or retrieval-based. We further validate different training strategies only for hard negatives in the ablation experiments.

## 4 EXPERIMENTS

To quantitatively evaluate the effectiveness of our MMGeoLM in geometric problem-solving, we conduct comparative experiments on four geometric benchmarks using various LMMs. Additionally, we analyze the impact of different vision encoder types, image-based negatives and training strategies in the subsequent ablation studies.

---

[1] `https://www.21cnjy.com/`

| Model | GeoQA | MathVista | | We-Math | | MM-Math | | | |
|---|---|---|---|---|---|---|---|---|---|
| | | GEO | ALG | S2 | S3 | Easy | Med | Hard | Avg |
| Human* | 92.3 | 48.4 | 50.9 | - | - | 90.7 | 81.9 | 47.6 | 80.4 |
| Claude-3-Opus* | 44.5 | 46.3 | 46.6 | 32.9 | 23.0 | 29.5 | 19.3 | 3.6 | 20.3 |
| Claude-3.5-Sonnet | 65.1 | 66.3 | 68.4 | **64.7** | **62.1** | 34.4 | 31.9 | 13.6 | 31.7 |
| GPT-4V* | - | 50.5 | 53.0 | 49.2 | 38.2 | 37.8 | 21.2 | 1.8 | 23.1 |
| GPT-4o | 58.9 | 62.7 | 65.4 | 58.0 | 43.6 | 45.8 | 30.0 | 10.9 | 31.8 |
| MAVIS-7B* | 68.3 | 64.1 | 59.2 | 37.9 | 34.6 | - | - | - | - |
| G-LLaVA-7B* | 67.0 | 56.7 | - | 30.1 | 32.7 | - | - | - | - |
| Math-LLaVA-13B | 48.4 | 55.6 | 55.0 | 31.7 | 23.0 | - | - | - | - |
| InternVL2-8B | 56.4 | 60.5 | 60.8 | 41.1 | 37.1 | 33.6 | 21.7 | 9.0 | 23.3 |
| LLaVA-OneVision-7B | 64.6 | 54.6 | 53.1 | 28.7 | 22.3 | 40.5 | 24.8 | 10.5 | 27.0 |
| Chimera-Reasoner-8B | **69.6** | 48.5 | 42.6 | 29.9 | 31.6 | 36.2 | 22.2 | 9.0 | 24.1 |
| Phi-3-Vision-128K-Instruct | 29.5 | 38.8 | 40.0 | 33.3 | 30.1 | 12.9 | 7.1 | 0.0 | 7.8 |
| Qwen2.5-VL-7B-Instruct | 59.0 | 67.7 | 66.9 | 58.1 | 50.6 | 50.7 | 32.2 | **14.2** | 34.8 |
| InternLM-XComposer2-7B | 38.8 | 63.0 | 56.6 | 33.1 | 33.0 | 18.9 | 12.2 | 4.5 | 13.1 |
| MMGeoLM-MAmmoTH2-7B | 68.5 | 68.7 | **69.5** | 41.5 | 37.5 | 52.5 | 32.4 | 4.5 | 36.9 |
| *w/o MMCLIP* | 55.4 | 52.2 | 51.3 | 34.1 | 35.3 | 42.5 | 21.8 | 4.5 | 26.8 |
| *Original AltCLIP* | 46.7 | 45.8 | 46.8 | 30.1 | 29.2 | 38.2 | 20.4 | 1.6 | 25.9 |
| MMGeoLM-Qwen2.5-7B | 69.2 | **69.8** | 68.1 | 39.8 | 36.4 | **55.3** | **36.9** | 9.0 | **39.2** |
| *w/o MMCLIP* | 56.3 | 54.1 | 50.3 | 35.2 | 35.1 | 50.5 | 28.8 | 4.5 | 34.3 |
| *Original AltCLIP* | 50.7 | 47.6 | 48.9 | 33.1 | 32.7 | 52.5 | 30.8 | 4.5 | 33.7 |

Table 2: Accuracy (%) of various LMMs on GeoQA, MathVista, We-Math, and MM-Math. 'S2/S3' denotes the two-step settings in We-Math. Results marked * are taken from their original papers.

## 4.1 MAIN EXPERIMENT: MMGEOLM EVALUATION

### 4.1.1 DATASETS AND METRICS

We assess the geometric reasoning capabilities of various LMMs across two types of benchmarks: multiple-choice and open-ended. The multiple-choice benchmark includes GeoQA (Chen et al., 2021), a geometric question answering task based on plane geometry; We-Math (Qiao et al., 2024), a visual mathematical reasoning task with questions of varying difficulty, solvable in two or three steps; and MathVista (Lu et al., 2023), widely used for evaluating LMMs' performance. The open-ended benchmark is MM-Math (Sun et al., 2024), which features high discriminative difficulty sourced from secondary school-level problems. We use accuracy (ACC) as the metric. For We-Math, we assess multi-step problems (S2 and S3), while for MM-Math, we evaluate models across easy, medium, and hard categories using a test set of 700 problems. For MathVista, we evaluate geometry problem-solving (GEO) and algebraic (ALG) categories.

### 4.1.2 BASELINES

The evaluated LMMs are categorized into two groups: Closed-source APIs: Claude-3-Opus (Anthropic, 2024), Claude-3.5-Sonnet, GPT-4o-20240513 (OpenAI, 2024a), and GPT-4V. Open-source LMMs: MAVIS-7B (Zhang et al., 2024c), G-LLaVA-7B (Gao et al., 2023), InternVL2-8B (Chen et al., 2024), LLaVA-OneVision-7B (Li et al., 2024), Qwen2.5-VL-7B-Instruct (Bai et al., 2025), Chimera-Reasoner-8B (Peng et al., 2024b), InternLM-XComposer2-7B (Dong et al., 2024), Phi-3-Vision-128K-Instruct (Abdin et al., 2024), and Math-LLaVA-13B (Shi et al., 2024). For the open-source category, we choose LMMs that have achieved strong results on geometric benchmarks (Chen et al., 2021; Lu et al., 2023). Additionally, we include human evaluation baselines, using scores from prior studies (Zhang et al., 2024c; Sun et al., 2024).

### 4.1.3 OVERALL RESULTS

As shown in Table 2, our proposed MMGeoLM achieves state-of-the-art performance on the Math-Vista and MM-Math benchmarks. On the GeoQA benchmark, MMGeoLM-Qwen2.5-7B lags

0.4% behind Chimera-Reasoner-8B. As Chimera-Reasoner-8B was trained on GeoQA (Peng et al., 2024b), MMGeoLM-Qwen2.5-7B achieves the best performance among other models that were not trained on this dataset. The improved performance in geometric problem-solving demonstrates the effectiveness of our training approach. For the We-Math benchmark, MMGeoLM-MAmmoTH2-7B underperforms GPT-4o, Claude-3.5-Sonnet, and the open-source Qwen2.5-VL-7B-Instruct model. We attribute this to We-Math's emphasis on recognition from visual elements rather than geometric mathematical reasoning, which limits the effectiveness of MMGeoLM-MAmmoTH2-7B's geometric reasoning enhancement strategies. On the open-ended MM-Math benchmark, MMGeoLM-Qwen2.5-7B performs poorly on hard problems, with less than 10% accuracy, but achieves over 55% accuracy on easy problems. Easy problems require fewer reasoning steps, allowing geometric element recognition to significantly enhance accuracy. In contrast, hard problems involve multi-step reasoning, where its impact is more limited. Compared to using the *w/o MMCLIP* (In-batch trained AltCLIP with random negatives), the vision encoder trained with our constructed hard negative samples significantly enhances MMGeoLM's geometric understanding capabilities. To rule out the possibility that performance gains stem from training–test leakage, we perform a contamination analysis (Appendix E) and we confirm that the evaluation sets contain no near-duplicate items.

## 4.2 ABLATIONS

### 4.2.1 ABLATION STUDY I: DIFFERENT NEGATIVE TYPES

| Vision Encoder | #Num. | GeoQA | MM-MATH | | | |
|---|---|---|---|---|---|---|
| | | | Easy | Med | Hard | Avg |
| Original | - | 45.4 | 40.5 | 20.8 | 4.5 | 23.8 |
| $Random_{10}$ | 400K | 45.0 | 39.8 | 22.7 | 4.5 | 24.9 |
| $Retrieval_{10}$ | 100K | 53.0 | 43.6 | 24.8 | 4.5 | 26.6 |
| $Rule_{10}$ | 100K | 48.8 | 45.4 | 27.6 | 4.5 | 28.1 |
| $Image\text{-}based_{10}$ | 4K | 54.9 | 45.6 | 26.5 | 4.5 | 29.0 |
| $Retrieval_{10}+Rule_{10}$ | 200K | 56.5 | 47.5 | 28.3 | 4.5 | 29.4 |
| **All Negatives** | 204K | **58.2** | **49.8** | **29.1** | 4.5 | **30.4** |

Table 3: Ablation study on vision encoder training with different types and quantities of hard negatives. The bottom panel reports performance when combining multiple types.

To further evaluate the performance of vision encoders trained with different types of hard negatives, we compare seven negatives summarized in Table 3. All models are based on AltCLIP. The Original vision encoder is directly initialized without any additional training, while $Random_{10}$ follows the conventional in-batch random negatives sampling strategy in Section 3.3. The remaining five vision encoders are trained with our proposed MMCLIP method in Section 3.3. The subscript '10' indicates a 1:10 positive-to-negative ratio. We further complement this analysis with retrieval evaluations reported in Appendix C, using the Hit@1 metric on four constructed evaluation sets.

**Results**. As shown in Table 3, vision encoders trained with hard negatives consistently outperform random in-batch negatives. even with only 4K image-based negatives , the $Image\text{-}based_{10}$ encoder achieves the highest average accuracy (29.0%) among all single hard negative methods on MM-Math. Furthermore, while $Retrieval_{10}$ and $Rule_{10}$ negatives individually provide moderate gains, their combination further improves performance (29.4%), indicating their complementary nature. The best results of All Negatives are achieved when all three types of negatives are combined, demonstrating that diverse hard negatives collectively enhance the model's geometric understanding. These findings underscore the importance of both the *type* and *diversity* of hard negatives in contrastive training for vision encoders.

### 4.2.2 ABLATION STUDY II: IMAGE-BASED NEGATIVES

**Code Perturbation vs. Rule-design: Assessing Image Negatives Construction Methods**. We contrast two image–based negative construction methods. PERTURB applies our code perturbation method to 4K images, generating negative images that preserve similar structure while introducing subtle differences. In contrast, $Rule_{img}$ is built from manually designed templates in the Mavis

| Method | #Num | GeoQA | MM-Math | | | |
|---|---|---|---|---|---|---|
| | | | Easy | Med | Hard | Avg |
| RULE$_{img}$ | 100K | 52.9 | 44.0 | 25.4 | 1.0 | 28.3 |
| PERTURB | 4K | 54.9 | 45.6 | 26.5 | 4.5 | 29.0 |
| $\Delta$ | **-96K** | **+2.0** | **+1.6** | **+1.1** | **+3.5** | **+0.7** |

Table 4: Effect of different image negative construction strategies. $\Delta$ rows give absolute gains of PERTURB over Rule$_{img}$.

Figure 4: An example of vision encoder training data with or without numeric markings.

| Visual Marking | GeoQA | MM-MATH | | | |
|---|---|---|---|---|---|
| | | Easy | Med | Hard | Avg |
| **w/** length/angle/etc. | 54.9 | 45.6 | 26.5 | 4.5 | 29.0 |
| **w/o** length/angle/etc. | 51.7 | 50.0 | 25.1 | 4.5 | 28.6 |

Table 5: Impact of numeric markings on vision encoder training. **w/** and **w/o** denote settings with and without such numeric annotations, respectively.

| In-Batch Negative | GeoQA | MM-MATH | | | |
|---|---|---|---|---|---|
| | | Easy | Med | Hard | Avg |
| Random | 46.2 | 41.8 | 23.7 | 4.5 | 25.5 |
| Image-based NS | **57.3** | **51.7** | **29.0** | 4.5 | **32.1** |

Table 6: Comparison of in-batch training with random negatives vs. in-batch training with image-based negatives. Mean score over 5 random seeds on 4K image-based negatives.

corpus. We construct 100K samples and retrieve the most similar images as hard negatives. To ensure a fair comparison, we fix the positive-to-negative ratio at 1:10 in both settings. Results in Table 4 show that the 4K PERTURB setting exceeds the 100K Rule$_{img}$ on both GeoQA (54.9 vs. 52.9) and MM-Math (29.0 vs. 28.3). These results highlight that code-perturbed negatives, despite being smaller in scale, outperform synthetic data due to their semantic diversity and closer resemblance to real-world tasks. This advantage is corroborated by additional analyses in Appendix D, where PERTURB-based negatives are shown to have a more distinct distribution than Rule$_{img}$ counterparts and higher semantic similarity to real-world supervised data.

**Length/Angle Marked Matters? Cross-Modal Consistency Analysis**. For a large amount of geometric problems, the conditions (lengths, angles, etc.) are provided only in textual format without corresponding image markings. This inherent asymmetry between image and text may lead to cross-modal hallucinations (vision encoder training vs. LMMs training). To assess these inconsistencies, we remove all numeric markings (lengths, angles, etc.) from the image-based negatives, retaining only letter labels (see Figure 4) for the vision encoder. Results in Table 5 show that without numeric markings leads to a slight performance drop (e.g., 29.0% → 28.6% on Avg, 54.9%→ 51.7% on GeoQA), suggesting that numeric markings have a moderate but non-negligible impact on alignment learning. This shows that the model primarily learns to align geometric features with text through structural cues, while numeric markings serve as auxiliary signals rather than essential supervision.

**Hard Negative Performance Gains or Trade-off for In-Batch Training**. We leverage the proposed image-based negative strategy for the in-batch training to verify the effectiveness. In the in-batch negatives setting, each sample in the batch serves as a hard negative for others, with traditional in-batch typically selected at random. For a fair comparison, both methods use 10 negative examples per positive sample. As shown in Table 6, in-batch training with our proposed image-based negative strategy outperforms In-Batch on the *easy*, *medium*, and *average* subsets under the same amount of training data. This highlights the advantage of explicitly constructed image-based negatives in guiding the model to better distinguish visually similar examples, thus achieving higher overall accuracy. Furthermore, these results indicate that carefully selected mutual hard negatives within batches lead to more effective contrastive learning.

### 4.2.3 ABLATION STUDY III: TRAINING STRATEGY AND ROBUSTNESS

**Data Scaling: More Hard Negatives Are Not Always Better**. Our proposed MMGeoLM method allows flexible scaling of negatives. However, whether performance improves proportionally with an increasing number of hard negatives remains unclear. To address this, we investigate the impact of varying the number of retrieval-based negatives per sample from 5 to 50. As shown in Figure 5, MMGeoLM's performance on MM-Math improves with increasing hard negative samples from 5 to

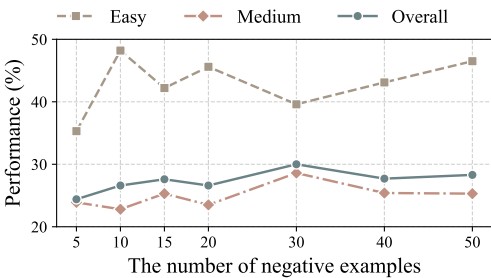

Figure 5: MMGeoLM performance with varying numbers of hard negative ratio.

Table 7: Robustness evaluation of MMGeoLM trained on text/image negatives under image transformers (rotation and downscaling).

| Transformers | Negatives | MM-MATH | | | |
|---|---|---|---|---|---|
| | | Easy | Med | Hard | Avg |
| No-Changing | Text-based | 48.6 | 29.1 | 9.0 | 31.9 |
| Rotate30° | Text-based | 47.4 | 28.5 | 9.0 | 31.0 |
| Downscaling | Text-based | 45.6 | 26.8 | 0.0 | 28.4 |
| No-Changing | Image-based | 45.6 | 26.5 | 4.5 | 29.0 |
| Rotate30° | Image-based | 50.0 | 25.6 | 4.5 | 29.0 |
| Downscaling | Image-based | 48.2 | 25.6 | 4.5 | 28.7 |

30, but slightly declines beyond this point. These results indicate that hard negative samples have diminishing returns beyond a certain threshold, with excessive examples reducing model performance.

**Robustness to Geometric Image Transformations**. Diagrams in real exams often undergo transformations such as rotation or scaling. To evaluate MMGeoLM's robustness, we apply two image perturbations: (1) a 30° clockwise rotation and (2) downscaling by 0.5. Table 7 reports the results under both *text-based* and *image-based* negative conditions. Model performance remains largely stable under a 30° rotation, showing only a minor accuracy drop of 0.9% in the text-based setting. This suggests the model is robust against moderate image rotations. In contrast, downscaling significantly impacts accuracy, dropping text-based performance from 31.9% to 28.4%. This decline indicates that textual captions are particularly sensitive to reduced image resolution whereas the image-based negative demonstrates stronger robustness, benefiting from diagram scale variations.

## 5 RELATED WORK

Mathematics-related research has recently received significant attention in large models. For text-only mathematical reasoning, several works employ external tools like Tora (Gou et al., 2023) or intermediate step decomposition methods such as MAmmoTH (Yue et al., 2023), Metamath (Yu et al., 2023) and Math-Shepherd (Yue et al., 2023). Many multimodal mathematics benchmarks have been proposed, including mathematical competition-oriented OlympiadBench (He et al., 2024), geometry-focused datasets like Geometry3K (Lu et al., 2021), VisScience (Jiang et al., 2024), Uni-Geo (Chen et al., 2022a) and GeoQA (Chen et al., 2022a). Many works take effort on training multi-modal math, including G-llava (Gao et al., 2023), Meta-LLaVA (Shi et al., 2024) and MAVIS (Zhang et al., 2024c). These models often struggle with fine-grained recognition of geometric elements. Vision encoders are critical for capturing spatial and structural information in multimodal tasks. Prior works (Zhang et al., 2024a; Doveh et al., 2023a;b; Singh et al., 2023) enhance image understanding using negative captions, but rarely target fine-grained geometric recognition. Methods such as Neg-CLIP (Yuksekgonul et al., 2022) and TriCLIP (Yang et al., 2024) focus on learning from negative samples, yet their effectiveness is limited by small or randomly selected negatives. In this work, we propose a scalable and geometry-aware negative learning strategy that explicitly constructs task-relevant hard negatives to improve geometric understanding.

## 6 CONCLUSION

This paper introduces two types of hard negatives, image-based and text-based negatives, targeted at geometric element understanding to enhance LMMs' geometric reasoning. Our resulting model, MMGeoLM, significantly outperformed existing models, even surpassing GPT-4o on key benchmarks. Our ablation analyses corroborate three key findings. First, both image-based and text-based negatives enhance geometric problem-solving, and their effects are complementary when used jointly. Second, the proposed image-based negatives, constructed via code perturbation, exhibit advantages in terms of data efficiency, annotation robustness, and enhanced performance combined with traditional in-batch training methods. Third, increasing the number of negatives does not monotonically improve performance; instead, our MMCLIP training strategy demonstrates robustness by balancing the quality and diversity of negatives, yielding consistent gains across benchmarks.

## ETHICS STATEMENT

Our 4K image-based negatives and 17K supervised fine-tuning training data were collected from the `21st Century Education`. We have obtained an official authorization agreement from them for research.

## REPRODUCIBILITY STATEMENT

The training dataset, 4K image-based negatives and 17K supervised fine-tuning training data, will be publicly released after the paper publication, with a download link to be provided in the camera-ready version. For the reproducibility of our training strategy, we provide three components. First, the implementation of MMCLIP training with both image-based and text-based negatives, along with the corresponding hyperparameters, is included in the supplementary material. Second, the MLP training setup follows the official implementation released in the LLaVA GitHub repository. Third, for incorporating LLMs, we adopt the training code and parameters from the official LLaVA GitHub repository for MAmmoTH2-7B, and we use the official code and configurations provided in the LLaVA-Next GitHub repository for Qwen2.5-7B.

## THE USE OF LARGE LANGUAGE MODELS

In this work, large language models were employed in three ways. First, they were used to generate code-based geometric diagrams and the corresponding captions for constructing image-based negatives. Second, they were applied to modify positive captions in order to derive negative captions for text-based negatives. Third, they were utilized to assist in polishing the writing of the manuscript.

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

## A    HARD NEGATIVE STATISTICS

We conduct a detailed analysis of the constructed hard negative datasets, covering the data sources, construction methods, the number of positive samples, the number of hard negatives per positive sample, and additional notes on the perturbation strategies. The statistics are summarized in Table 8.

| Data Type | Source & Method | #Positives | #Hard Negatives | Notes |
|---|---|---|---|---|
| Retrieval-based Negative | MAVIS + SimANS | 100K | 10-50 | Dense retrieval, high semantic similarity |
| Rule-based Negative | MAVIS + GLM-4 | 100K | 10 | Shape / relation / value perturbations |
| Image-based Negative | New Collection + Gemini2.5 | 4K | 10 | Code-driven geometric modifications |

Table 8: Statistics of constructed hard-negative samples. "#Hard Negatives" indicates the number of negatives per positive sample.

## B    EXAMPLES OF IMAGE-BASED NEGATIVES

Figure 6 presents some examples of image-based negatives, covering analytical geometry and planar geometry. Geometric elements include triangles, quadrilaterals, and various relational properties such as perpendicularity and intersection. Additionally, certain length/angle properties absent from the original diagrams are explicitly indicated. Specifically, the first column shows the original geometric images, and the second column depicts images generated from the given questions and the true answer, which, despite slight discrepancies, accurately capture the overall outlines of the original diagrams. The third and fourth columns display the constructed hard negative image based upon their corresponding negative captions and corresponding Python script—demonstrating notable similarities but clear and meaningful distinctions.

## C    RETRIEVAL EXPERIMENTS FOR VISION ENCODER

In addition to ablation evaluation, retrieval performance offers a direct measure of image–text alignment, computed via similarity scores between paired inputs. As a result, the ranking position of the ground-truth positive sample reflects the quality of learned representations. To assess retrieval capabilities, we evaluate the trained AltCLIP models using the *Hit@1* metric, which measures whether the correct caption is ranked highest among a large pool of candidates. Specifically, we construct four evaluation sets of hard negatives, each containing 500 samples, corresponding to the following strategies:

- **Random Negative**: Caption negatives randomly selected.
- **Retrieval Negative**: Caption negatives retrieved using the SimANS model, selecting the top 100 most similar captions to the positive sample.
- **Rule-based Negative**: Caption negatives generated according to the rules defined in earlier section.
- **Image-based Negative**: Image negatives generated from real-world geometric problems from new collection.

Table 9 reports retrieval performance of various vision encoders trained with different types of negative samples in this experiment.

The retrieval performance of different AltCLIP variants is summarized in Figure 7. Under the image-based negative setting, AltCLIP-Image achieves the highest *Hit@1* score (23%), substantially outperforming all other variants, most of which remain below 15%. This demonstrates the superior alignment learned from visually grounded negatives. However, the performance is still constrained, likely due to the limited size of the image-based dataset (4K samples), which may restrict the model's capacity to generalize under this setting. In retrieval- and rule-based evaluations, AltCLIP-Rand achieves the highest scores among single-source variants (64% and 31%, respectively), outperforming both AltCLIP-Retrieval and AltCLIP-Rule. This suggests that while domain-specific negatives help the model specialize, pre-training on random negatives provides broader generalization capabilities. Combining random negatives with domain-specific ones further improves robustness.

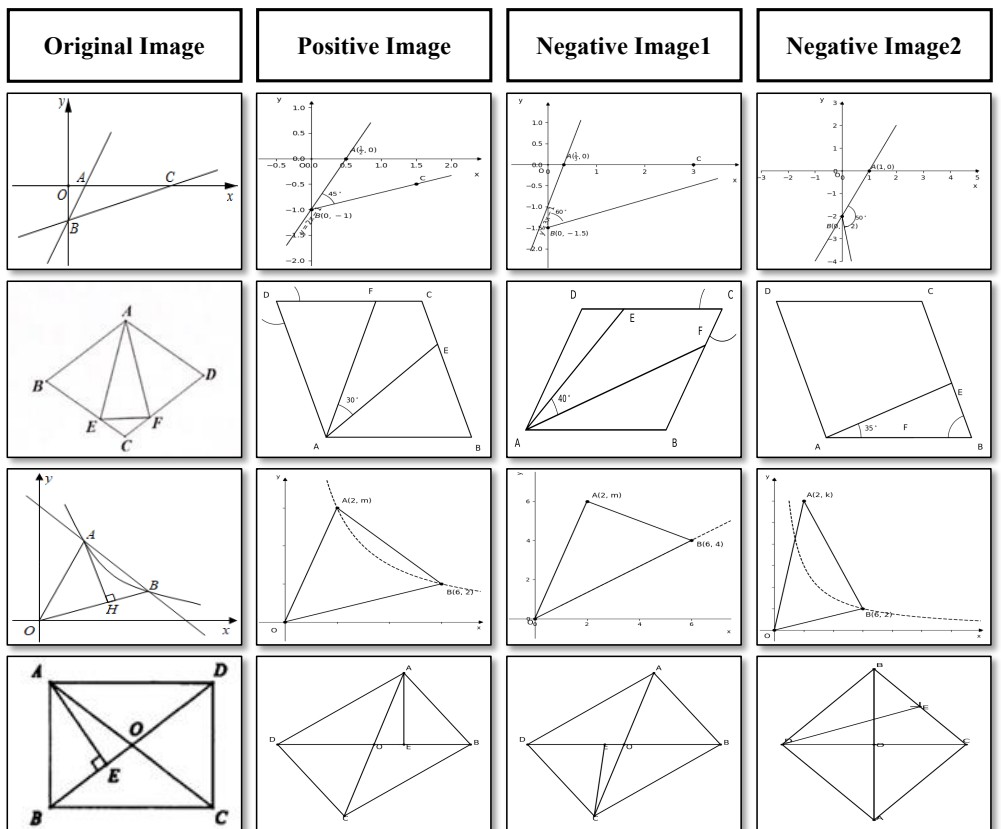

Figure 6: Comparison of generated geometric figures. The first column shows the original images, the second column presents the positive images, and the last two columns illustrate the constructed hard negatives.

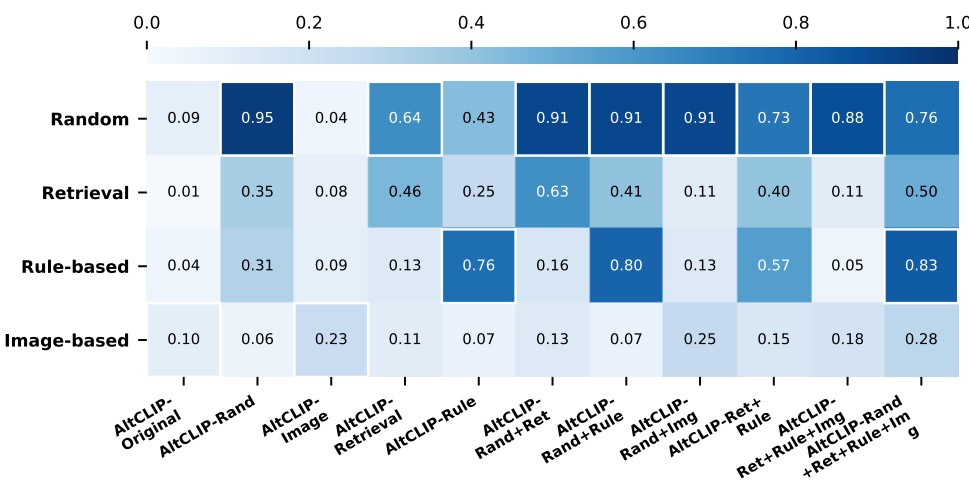

Figure 7: Retrieval performance of AltCLIP variants trained with different strategies.

For instance, AltCLIP-Rand+Rule yields strong performance across retrieval (41%) and rule-based (80%) settings, while AltCLIP-Rand+Ret+Rule+Img consistently achieves top scores across all four settings. These results indicate that starting with random negatives and progressively incorporating

| AltCLIP Variant | #Negative Source (K) | | | | Neg pos | Train |
|---|---|---|---|---|---|---|
| | RND | RET | RULE | IMG | | |
| AltCLIP-Original | – | – | – | – | – | – |
| AltCLIP-Rand | 400 | – | – | – | 10 | IB |
| AltCLIP-Retrieval | – | 100 | – | – | 10 | MM |
| AltCLIP-Rule | – | – | 100 | – | 10 | MM |
| AltCLIP-Image | – | – | – | 4 | 10 | MM |
| AltCLIP-Rand+Ret | 400 | 100 | – | – | 10 | IB+MM |
| AltCLIP-Rand+Rule | 400 | – | 100 | – | 10 | IB+MM |
| AltCLIP-Rand+Img | 400 | – | – | 4 | 10 | IB+MM |
| AltCLIP-Ret+Rule | – | 100 | 100 | – | 10 | MM |
| AltCLIP-Ret+Rule+Img | – | 100 | 100 | 4 | 10 | MM |
| AltCLIP-All | 400 | 100 | 100 | 4 | 10 | IB+MM |

Table 9: Dataset composition for each **AltCLIP** vision encoder variant. Columns list the number of positive pairs (in thousands) drawn from four negative-source types: **RND** = random, **RET** = retrieval, **RULE** = rule-based, **IMG** = image-based. *Neg/pos* indicates negatives-per-positive; training strategies: **IB** = In-batch, **MM** = MMCLIP, **IB+MM** = hybrid.

| Negative Type | Max Cosine Similarity | | | |
|---|---|---|---|---|
| | $> 0.90$ | $> 0.85$ | $> 0.70$ | $> 0.60$ |
| Image-based *vs.* SFT | 20.89% | 78.68% | 98.43% | 99.43% |
| Retrieval-based *vs.* SFT | 0.00% | 26.18% | 98.00% | 99.28% |

Table 10: Semantic similarity between hard negative captions and the 17K SFT dataset. Values denote the percentage of negative captions whose *maximum* cosine similarity with any SFT question exceeds the specified threshold.

hard negatives can effectively balance generalization and domain-specific precision in image–text alignment.

## D WHY FEWER SHOT IMAGE-BASED NEGATIVES WORK: A SIMILARITY AND DISTRIBUTION STUDY

In above ablation study, we observe that only 4K image-based negatives generated can match or even surpass the performance of 100K retrieval-based negative image from MAVIS. To better understand this phenomenon, we conduct two follow-up analyses.

First, we evaluate the semantic similarity between negative samples and the supervised fine-tuning dataset. Specifically, we employ the sup-simcse-roberta-large model to encode (1) the captions of image-based negatives, (2) the top-10 retrieval-based negative captions, and (3) the questions from the 17K SFT dataset. We then compute the cosine similarity between each caption and the questions in SFT, and for each caption, we retain the maximum similarity score as its semantic alignment score. Next, we calculate the proportion of captions whose maximum similarity exceeds various thresholds (0.90, 0.85, 0.70, and 0.60). As shown in Table 10, image-based negatives exhibit substantially higher similarity to the SFT dataset, with 20.89% exceeding 0.90 and 78.68% exceeding 0.85. In contrast, retrieval-based negatives show much lower proportions, with only 0.00% and 26.18% exceeding these respective thresholds. These results suggest that image-based negatives, which are derived from real-world problems, exhibit closer semantic alignment with downstream supervised data compared to artificially constructed samples.

Next, we visualized the embeddings of both image-based negatives and retrieval-based negatives using t-SNE, reducing them to two dimensions, as shown in Figure 8. The visualization clearly shows a distinct separation between the two types of negative samples. We further evaluate the separation of the two negative types using K-means clustering on the embeddings of both image-based and retrieval-based negatives. We then calculated two separation metrics: K-means Separation Accuracy and Natural Separation Score. The results are as Table 11:

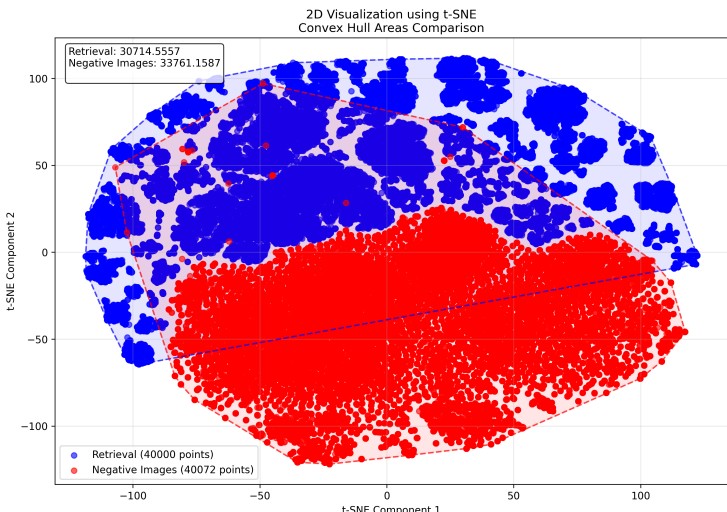

Figure 8: Comparison of GPT-4o and MMGeoLM in geometric problem-solving. Both models produce incorrect answers, but MMGeoLM's solution is closer to the True Answer.

| Metric | Value |
|---|---|
| K-means Separation Accuracy | 0.90 |
| Natural Separation Score | 0.90 |

Table 11: Clustering-based separation metrics between image-based and retrieval negative samples. Higher values indicate stronger distinguishability between the two groups in embedding space.

From Table 11, we observe that both the K-means Separation Accuracy and the Natural Separation Score reach 0.90, indicating a strong degree of separability between the two types of negatives. This suggests that the 4K image-based negatives are not only distinguishable from retrieval-based ones, but also semantically closer to downstream tasks. Consequently, their effectiveness stems from their higher alignment with real-world data, rather than relying on large-scale rule-based construction. In contrast, datasets built purely through human-making rules may require a substantially larger volume to achieve comparable performance, resulting in lower data efficiency.

## E TRAINING-TEST OVERLAP AND DATA-CONTAMINATION ANALYSIS

Both the training and test datasets are drawn from genuine exam questions in an open-ended format; thus, accidental duplicates could harm evaluation scores. To assess this risk, we embed every question with *sup-simcse-roberta-large* and compute the cosine similarity between each test question and the entire training datasets. For each test question we record its *maximum* and *top-5* similarity scores.

Figure 9 shows the examples of top 5 similar score pair . For the top 1 score (*sim*= 0.9977), although their wording is close, a key numerical change ($AB = 5, DC = 11$ vs. $AB = 6, DC = 13$) yields a absolute different solution path. To ensure a clean separation between training and test sets, we remove from the training data any question whose similarity with any test question exceeds 0.995. Table 12 summarizes the resulting overlap statistics. Under this filtering strategy, **100%** of the test questions have no near-duplicate counterparts in the training data, and over **97%** have a maximum similarity below 0.990, demonstrating that our evaluation is not compromised by training leakage.

## F HUMAN VERIFICATION OF GENERATED CAPTIONS

We randomly sampled 123 captions generated by the LLM call from the pipeline of image-based negatives. The sample size was chosen to cover diverse geometric configurations (triangles, quadri-

**Test Question:**
As shown in the figure, in quadrilateral ABCD, AB ∥CD, AB = 5, DC = 11, and the sum of AD and BC is 12. Points E, F, and G are the midpoints of BD, AC, and DC respectively. What is the perimeter of △ EFG?

**- Top 5 Matches:**

**1** **Train Question:**
As shown in the figure, in the quadrilateral ABCD, AB ∥CD, AB = 6, DC = 13, and the sum of AD and BC is 12. Points E, F, and G are the midpoints of BD, AC, and DC respectively. What is the perimeter of △EFG?
*Similarity*: 0.9977

**2** **Train Question:**
As shown in the figure, in △ABC, AB = 3, AC = 5, and points D, E, and F are the midpoints of AB, BC, and AC respectively. What is the perimeter of quadrilateral ADEF?
*Similarity*: 0.9706

**3** **Train Question:**
As shown in the figure, in △ABC, AB=10, AC=7, BC=9, and points D, E, and F are the midpoints of AB, AC, and BC, respectively. What is the perimeter of the quadrilateral DBFE?
*Similarity*: 0.9654

**4** **Train Question:**
As shown in the figure, in △ABC, D, E, and F are the midpoints of AB, BC, and CA, respectively, with AC=20 and BC=18. What is the perimeter of the quadrilateral DECF?
*Similarity*: 0.9620

**5** **Train Question:**
As shown in the figure, in △ABC, points D, E, and F are the midpoints of AC, BC, and AB respectively. Given that AB = 12, BC = 8, and DE = 4.5, what is the perimeter of △DEF?
*Similarity*: 0.9594

Figure 9: Examples of top-5 most similar training–test question pairs used for contamination analysis. Each pair includes the test question, its top training matches, and the associated cosine similarity.

| Similarity Threshold | Percentage |
|---|---|
| < 0.995 | 100% |
| < 0.990 | 97.28% |
| < 0.980 | 90.56% |
| < 0.970 | 81.26% |

Table 12: Proportion of test questions whose maximum similarity to the training set falls below various thresholds, after filtering.

laterals, circles, etc.). Each caption was independently evaluated by three annotators with a background in mathematics education at the college level. Annotators were asked to check whether every geometric element described in the caption (e.g., points, lines, angles, shapes) was consistent with the diagram rendered from code. The task was framed as a binary decision: correct (all elements accurately represented) vs. incorrect (at least one element misrepresented or missing). All three annotators agreed on every instance, yielding 100% accuracy with full inter-annotator agreement. Although the agreement was perfect on this sample, the evaluation was limited to 123 captions. Larger-scale or more diverse verification may reveal edge cases where captions misrepresent subtle geometric relations.

## G PROMPTS

After multiple rounds of experimentation, we finalized six prompts for our data construction, as illustrated in Figures 10 to 15. Specifically, the prompt in Figure 10 input a textual geometry problem along with its answer and generates a corresponding Python script that renders the geometric image. Based on this script, we further generate a positive caption using the prompt shown in Figure 11. Directly perturbing the Python script alone to get image often leads to uncontrolled. Therefore,

as shown in Figure 12, we first generate negative captions by introducing subtle but semantically significant modifications to the positive caption. We then use these negative captions to perturb the original Python script, ultimately producing visually plausible but negative images, as shown in Figure 13. Since the images are programmatically generated, errors in the scripts are sometimes inevitable. To ensure quality, we employ a LLM to automatically verify and correct any syntactic or semantic issues in the generated code by Figure 14. For rule-based text negatives, we design specialized prompts to enforce targeted modifications. The corresponding prompt is illustrated in Figure 15, which enables controlled perturbations based on observed reasoning errors.

---

**Given a geometry problem and its answer, generate a Python script to accurately render the corresponding geometric figure. The following requirements must be strictly followed:**
**1. Use only black color for all graphical elements; do not use any colored elements.**
**2. Do not include any title in the figure.**
**3. Ignore any auxiliary or extended lines that may be relevant for reasoning but are not explicitly part of the original figure.**
**4. Only annotate the geometric elements (such as coordinates, lengths, and angles) explicitly mentioned in the problem statement. Do not include any additional annotations derived from the answer.**
**5. For analytic geometry problems, annotate known equations if provided in the problem.**
**6. Ensure that all geometric elements are displayed within the boundaries of the figure; do not place elements outside the visible area.**
**7. Save the final figure as an image file named `question.png`.**
**Question: {}**
**True Answer: {}**
**Output Python script :**

Figure 10: Prompt for generating a geometric Python script from a given geometry problem and its solution. The resulting script is used to render the corresponding image.

---

**Given a Python script that generates a geometric diagram, read the code and produce a textual description of the diagram according to the following rules:**
**1. Ensure accurate descriptions of geometric elements in the diagram.**
**2. Do not mention formatting details such as black dots, solid black lines, or the presence or absence of axis ticks.**
**3. Include necessary mathematical symbols such as $\triangle$ and $\angle$ instead of using plain text like "triangle" or "angle".**
**4. Avoid including inferred calculations; instead, directly describe the results of such reasoning as part of the description. For example, write "$\angle AOM = 10°$" instead of "$\angle AOM = \angle AOC - \angle MOC = 30° - 20° = 10°$". However, only include such a result if $\angle AOM$ is explicitly labeled in the diagram.**
**5. Do not include introductory phrases such as "The following is a textual description of the geometric diagram generated by the provided Python code."**

**Python script: {}**

Figure 11: Prompt for generating a positive geometric caption from a Python script.

## H    CASE STUDY OF PROBLEM SOLVING

In Figures 16 and 17, we compare the solutions generated by GPT-4o and our MMGeoLM. In Figure 16, the red text highlights GPT-4o's misrecognition of two triangles in the image, leading to an incorrect final result. In contrast, MMGeoLM produces a correct solution, albeit through a reasoning process different from the ground-truth answer. This highlights MMGeoLM's capability to generate diverse problem-solving approaches. Figure 17 illustrates errors in image-based reasoning for

Given a positive caption describing a geometric figure and its corresponding Python code, generate 10 perturbed negative caption. These negative samples are intended to support contrastive learning by introducing subtle inconsistencies that challenge the model's image-text alignment ability. The perturbations must maintain structural plausibility while differing from the original in meaningful ways. Follow these constraints:

1. Modify geometric attributes such as shape types (e.g., change "square ABCD" to "rectangle ABCD"), curve properties (e.g., change "diameter of circle o1" to "diameter of ellipse o1"), or triangle types (e.g., change "equilateral triangle" to "isosceles triangle").
2. Modify letter-based indicators by changing at least two letters and avoid permutations that preserve original start and end sequences. For instance, changing "ABCD" to "CBDA" is acceptable, but "ABCD" to "CDAB" or "AC" to "CA" is not.
3. Adjust numerical values (e.g., change "the side length of square ABCD is 8.0" to "6.0").
4. Substitute verbs with similar semantics where appropriate (e.g., "is similar to" vs. "is congruent to"), but avoid drastic changes like replacing "parallel" with "perpendicular," which would distort the image meaning.
5. Changing geometric annotations (e.g., replacing "side AB" with "segment BA") is considered incorrect and should be avoided, as these refer to equivalent constructs in geometry.
6. Do not introduce letters or elements that do not appear in the original figure (e.g., changing "trapezoid GHIJ" to "trapezoid XYZO" is invalid).
7. Distribute the modifications across the entire caption. Avoid concentrating all changes in only the beginning or the end.

Positive Caption: {}
Python Code: {}

Figure 12: Prompt for generating negative captions by introducing fine-grained semantic modifications to the positive caption.

Given a Python script that draws a geometric diagram and a negative-sample caption describing the desired diagram, modify the script so that it satisfies the caption, following these rules:
1. The final diagram must match the negative-sample caption and be saved as "question.png".
2. Preserve the original coding style and formatting conventions of the geometric drawing.
3. Ensure that every geometric element stays within the figure boundaries.
4. Remove any call to plt.show(); instead, save the figure as "question.png".

Negative caption description: {}
Python script: {}

Figure 13: Prompt for generating negative geometric images by modifying the original Python script based on the perturbed negative captions.

Given a Python script and its corresponding error message, revise the code according to the following rules:
1. Inspect the code for errors; if any are found, correct them.
2. Verify that the final figure is saved as a PNG file in the same directory as the script; add the save command if it is missing.
3. Remove any calls to plt.show().
4. Return only the corrected code, with no additional text.

Python script: {}
Error message: {}

Figure 14: Prompt for automatic correction of syntactic or semantic errors in Python scripts using an LLM. This ensures that all generated diagrams are valid and executable.

> **Given a positive-sample caption that describes a geometric diagram, generate 10 perturbed captions, each containing 1 to 4 edits. Follow these rules:**
>
> **1. Types of allowable edits**
>
> • **Change properties of geometric objects** (e.g., replace "square ABCD" with "rectangle ABCD"; change "circle $o_1$ diameter" to "circle $o_1$ radius"; change "equilateral $\triangle ABC$" to "isosceles $\triangle ABC$").
>
> • **Re-order or substitute letter labels.** When modifying labels, alter at least two letters and avoid keeping the same leading or trailing sequence (e.g., change "square ABCD" to "square CBDA"; change "segment AB ∥ CD" to "segment AD ∥ CB").
>
> • **Adjust numeric values** (e.g., replace "side length 8.0" with "side length 6.0").
>
> • **Replace transformation verbs** (e.g., change "segment AB ∥ CD" to "segment AB ⊥ CD"; change "similar" to "congruent", "equal" to "parallel", etc.).
>
> **2. Constraints on edits**
>
> • Do **not** introduce letters that do not appear anywhere in the original caption (e.g., do not turn "trapezoid GHIJ" into "trapezoid XYZO" if X, Y, Z, O never appear in the diagram).
>
> • Distribute edits evenly throughout the caption; do not cluster all changes at the beginning or end. Output exactly 10 modified captions.
>
> **Positive-sample caption: {}**

Figure 15: Prompt for constructing rule-based text negatives through controlled perturbations of the original caption.

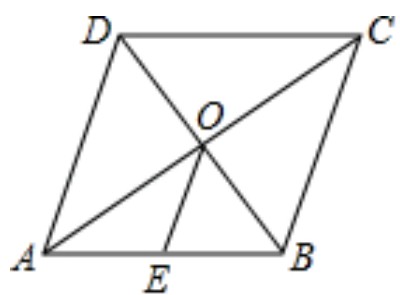

**GPT4o:** First, we analyze the properties of the rhombus ABCD. Since AC and BD are the diagonals of the rhombus, they are perpendicular to each other and intersect at their midpoint O. ,..., Since AC is the midline, it also divides △ ABD into two congruent triangles, △ AEO and △ BEO. In △ AEO, AE is half the length of AB, so AE $= \frac{1}{2}$ AB. ,...., $OE^2 = 4^2 - 2^2$, $OE^2 = 16 - 4$, $OE^2 = 12$, $OE = \sqrt{12}$, $OE = 2\sqrt{3}$, Thus, the final answer is $\boxed{2\sqrt{3}}$

**MMGeoLM:** ∵ Quadrilateral ABCD is a rhombus, AC=8, BD=6, ∴ AC ⊥ B, AO=CO=4, BO=DO=3, ∴ $AB = \sqrt{AO^2 + BO^2} = \sqrt{4^2 + 3^2} = 5$, ∵ Point E is the midpoint of AB, ∴ OE $= \frac{1}{2}$ AB $= \frac{1}{2} \times 5 = \boxed{2.5}$.

**Question:** As shown in the figure, in the rhombus ABCD, the diagonals AC and BD intersect at point O. Point E is the midpoint of AB, and it is given that AC = 8 and BD = 6. What is the length of OE?

**True Answer:** ∵Quadrilateral ABCD is a rhombus, ∴ $AC \perp BD$, and AO = CO, DO = BO. ∵AC = 8, BD = 6, ∴ $AO = \frac{1}{2}AC = \frac{1}{2} \times 8 = 4$, and $OD = \frac{1}{2}BD = \frac{1}{2} \times 6 = 3$. In △AOD, $AD = \sqrt{OD^2 + OA^2} = \sqrt{3^2 + 4^2} = 5$. Furthermore, ∵point E is the midpoint of AB, and O is the midpoint of BD, ∴ $OE = \frac{1}{2}AB = \frac{1}{2} \times 5 = \frac{5}{2} = \boxed{2.5}$.

Figure 16: Comparison of GPT-4o and MMGeoLM in geometric problem-solving. MMGeoLM correctly answers the problem using a different solution compared to True Answer, while GPT-4o generates a non-existent element $\triangle ABD$.

both MMGeoLM and GPT-4o. While both models make mistakes, MMGeoLM's reasoning aligns more closely with the correct solution, demonstrating improved mathematical reasoning. Overall, our trained MMGeoLM model demonstrates greater reasoning diversity and enhanced geometric problem-solving performance compared to current mainstream models.

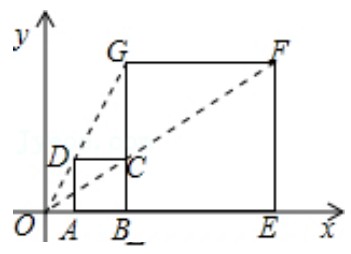

**Question:** As shown in the figure, in the Cartesian coordinate system, square ABCD and square BEFG are similar figures with the origin O as the center of similarity, and the similarity ratio is $\frac{1}{3}$. Points A, B, and E lie on the x-axis. If the side length of square BEFG is 6, what is the coordinate of point C?

**GPT4o**: First, since the similarity ratio between square BEFG and square ABCD is $\frac{1}{3}$, and the side length of square BEFG is 6, the side length of square ABCD is 6 ×3 = 18. Since points A, B, and E lie on the x-axis, and one side of square ABCD lies on the x-axis with point A coinciding with the origin O, the x-coordinate of point B is the length of AB, i.e., B(18, 0). ,…, so it is -18. Therefore, the coordinates of point C are (18, -18). Thus, the final answer is (18, −18).

**MMGeoLM**: Since the side length of square BEFG is 6, the coordinates of point E are (6, 0). Since the similarity ratio is 1:3, the coordinates of point B are (2, 0). Since square ABCD is similar to square BEFG, the coordinates of point C are (2, 6). Therefore, the answer is (2,6).

**True Answer**: ∵Square ABCD and square BEFG are similar figures with the origin O as the center of similarity, and the similarity ratio is $\frac{1}{3}$, ∴ $\frac{BC}{EF} = \frac{1}{3}$. ∵EF = 6, ∴BC = 2. ∵△ OBC ∼△ OEF, ∴ $\frac{OB}{OE} = \frac{BC}{EF}$. ∴ $\frac{OB}{OB+6} = \frac{1}{3}$, ∴OB = 3. ∴The coordinates of point C are (3,2).

Figure 17: Comparison of GPT-4o and MMGeoLM in geometric problem-solving. Both models produce incorrect answers, but MMGeoLM's solution is closer to the True Answer.

