# OpenReview forum: "MMGeoLM: Hard Negative Contrastive Learning for Fine-Grained Geometric Understanding in Large Multimodal Models"
_ICLR.cc/2026/Conference — Submitted to ICLR 2026_

### Official Review · Reviewer_q6Rm · 2025-10-28

**Soundness:** 2
**Presentation:** 2
**Contribution:** 2
**Rating:** 2
**Confidence:** 4

**Summary:**

This paper introduces MMGeoLM, a large multimodal model for geometric mathematical reasoning, built upon a novel hard negative contrastive learning framework (MMCLIP). The method enhances fine-grained geometric understanding by constructing two types of structured hard negatives: (1) text-based (retrieval- and rule-based) negatives that modify geometric relations or attributes, and (2) image-based negatives generated by perturbing LLM-produced diagram code. Trained with these negatives, the vision encoder significantly improves alignment quality, leading MMGeoLM to achieve state-of-the-art performance on four geometry reasoning benchmarks, even surpassing GPT-4o on MM-Math.

**Strengths:**

- This paper addresses the overlooked issue of geometric fine-grained perception in LMMs through a principled and generalizable hard negative framework.

- The code-perturbation strategy for generating image-based negatives is elegant, efficient, and produces semantically meaningful supervision.

- Strong performance across multiple benchmarks, detailed ablations, and robustness analyses clearly demonstrate the method’s effectiveness and scalability.

**Weaknesses:**

- The introduction and Figure 1 clearly diagnose the major issue of existing LMMs: a.hallucinating non-existent geometric elements or b. misinterpreting spatial relationships. However, the experiments only report overall accuracy without any qualitative or quantitative evaluation of whether MMGeoLM actually reduces such hallucinations. This leaves the key claim unverified.

- The evaluation omits several sub-benchmarks in MathVista, despite it being a comprehensive multimodal reasoning dataset. This selective testing weakens the evidence for broad generalization and comparability with prior works.

- Both training and evaluation focus narrowly on geometry-related math images. The paper does not explore whether the proposed hard negative framework benefits other structured visual domains—such as tables, charts, or diagrams—where fine-grained spatial reasoning is also required. This raises concerns about the model’s generalization beyond in-domain settings.

- The proposed hard-negative loss remains a variant of the standard InfoNCE formulation. The work does not explore introducing structural constraints (e.g., graph-based or relation-aware embeddings) to explicitly model geometric relationships, which could more directly address fine-grained spatial reasoning.

**Questions:**

same as weakness

---

> ### Author Response · Authors · 2025-11-25
>
> ## Q1. Evaluation of Hallucination Reduction
> Thank you for pointing this out.
>
> Following the automatic evaluation procedure from MM-Math [1], we analyze the first error in each answer and classify it into: (1) image misunderstanding, (2) reasoning error, (3) calculation error, and (4) question misunderstanding.
>
> | Model | Image | Reasoning | Calculation | Question |
> |-------|-------|-----------|-------------|----------|
> | GPT-4o | 56.6 | 36.4 | 3.4 | 3.6 |
> | MMGeoLM-Qwen2.5-7B | 28.3 (−28.3%) | 57.1 (+20.7%) | 8.6 | 6.0 |
>
> Image misunderstanding drops from 56.6% (GPT-4o) to 28.3% in MMGeoLM, which directly corresponds to hallucinating nonexistent geometric elements or misreading spatial relations.
>
> This provides quantitative evidence that MMGeoLM effectively reduces the hallucination issues described in the introduction and Figure 1.
>
> We also include qualitative examples in the Appendix  C (Retrieval Experiments for the Vision Encoder), showing that geometry-aware hard negatives improve image–text alignment and strengthen geometric element recognition.
>
> ## Q2. MathVista Sub-benchmarks
> Our work is designed for **geometric mathematical reasoning**, where fine-grained geometric variations can be reliably constructed through code-based perturbation. For this reason, we report results on the **GEO** and **ALG** subsets of MathVista, which directly evaluate geometry-aware reasoning.
>
> Other MathVista subsets mainly test general VQA skills that are orthogonal to our focus. Including them would dilute the core message.
>
>
> ## Q3. Generalization Beyond Geometry
> Exploring **broader domains (tables, charts, or diagrams)** is beyond the scope of the our work and would constitute a different problem setting (e.g., “hard negatives for structured visual diagrams”). We will clarify this limitation in the revision.
>
> ## Q4. On Structural Encoders (Graph- or Relation-Aware Representations)
> Methods that introduce explicit geometric structures have been explored in prior works including GeoDANO[1], GEOX[2], and MAVIS[3].
> These approaches are not the core novelty of our method, and we compare them in our experiments **(Sec. 4.2.2 “Hard Negative Performance Gains or Trade-offs for In-Batch Training” and Appendix D)**.
>
> [1] GeoDANO: Geometric VLM with Domain Agnostic Vision Encoder
>
> [2] GeoX: Geometric Problem Solving Through Unified Formalized Vision-Language Pre-training
>
> [3] MAVIS: Mathematical Visual Instruction Tuning with an Automatic Data Engine

---

### Official Review · Reviewer_xduf · 2025-10-30

**Soundness:** 2
**Presentation:** 3
**Contribution:** 2
**Rating:** 2
**Confidence:** 5

**Summary:**

This paper proposes MMCLIP, a hard negative contrastive learning framework designed to improve visual encoders for geometric reasoning. The method constructs both image-based hard negatives and text-based hard negatives, derived from rule-based modifications or caption retrieval. The trained encoder is then used to build MMGeoLM, a 7B LMM that shows improvements over open-source baselines on three geometric reasoning benchmarks.

**Strengths:**

The paper targets an important  challenge—enhancing geometry-specific visual encoders for reasoning tasks.

The proposed hard negative generation approach is conceptually sound and improves the visual encoder’s fine-grained discrimination.

**Weaknesses:**

The main contribution lies in applying hard negative contrastive learning to geometric visual encoders, which, while useful, represents an incremental improvement (more like technic report) rather than a fundamentally new research direction. The approach primarily involves building a domain-specific dataset, generating hard negatives, and training with standard architectures and loss functions, without methodological innovation in model design or learning objectives.

Moreover, geometry-specific vision encoders have already been explored in multiple prior works, including GEOX, GeoDANO, and Primitive Vision. These works similarly addressed fine-grained geometric perception using both image-level encoders (e.g., CLIP, SigLIP, MAE, VQ-GAN) and object-level detectors (e.g., Grounding DINO/Faster R-CNN).


**References**

[1]  GEOX: GEOMETRIC PROBLEM SOLVING THROUGH UNIFIED FORMALIZED VISION-LANGUAGE PRE- TRAINING

[2]  GeoDANO: Geometric VLM with Domain Agnostic Vision Encoder

[3]  Primitive Vision: Improving Diagram Understanding in MLLMs

**Questions:**

The experimental scope is also limited. The evaluation is conducted only on smaller-scale models (7B parameters), without examining performance scalability or testing on more complex benchmarks such as DynaMath, which assesses reasoning under visual dynamics and temporal changes.

---

> ### Author Response · Authors · 2025-11-16
>
> ## Q1 – On the claim that our contribution is merely incremental / a technical report
>
> Thank you for raising this concern. The novelty of our work lies in **automatically constructing high-quality geometry-aware code diagrams**, which are mainly used to generate image-based hard negatives in this work. These contributions, to the best of our knowledge, are not present in prior works. Moreover, recent “reason-while-drawing’’ models such as Zebra[1] and MathCanvas[2] further highlight the need for automatically constructing geometry-aware diagram code.
>
> ### A fully automated pipeline that constructs geometry diagrams directly from real exam problems, rather than from a small set of geometric primitives.
>
> Works such as MAVIS[3], GeoDANO[4], and other template-driven systems generate diagrams based on a limited set of primitive elements (points, lines, angles) or pre-defined geometric templates.
> They cannot capture the diversity found in real exam problems—for example, Pythagorean-based constructions or triangle-similarity reasoning.
> Our pipeline starts from **real exam questions** and more closely matches authentic problem structures.
>
> ### Automated generation of high-quality, geometry-faithful diagrams without human annotation.
>
> Benchmarks such as DynaMath[5] rely on human annotators to write, execute, and validate diagram-generating code.
> In contrast, our system uses LLMs to parse the problem, generate executable Python code, auto-correct syntax, and produce captions without any human annotation, while still achieving high geometric fidelity.
>
> ### Image-based hard negatives with subtle but critical geometric differences.
>
> Unlike diffusion-generated negatives (e.g., TripletCLIP[6]) or randomly chosen geometric images,
> our negatives are created by minimal structural perturbations (e.g., equilateral → isosceles),
> while still preserving the geometric constraints of the original diagram.
> Thus, the negative images are both plausible and minimally different—an essential property for hard negatives.
>
> ### Recent unified multimodal models require high-quality code-based geometry diagrams.
>
> As shown in Appendix Figure 6, our generated code-based diagrams maintain high fidelity to real exam problems (despite minor angle deviations).
> These high-quality diagrams are important for unified multimodal models that perform reason-while-drawing, such as Zebra[1] and MathCanvas[2], which depend on accurate and diverse code-based geometric diagrams.
>
> [1] Zebra: Zebra-CoT: A Dataset for Interleaved Vision Language Reasoning
>
> [2] MathCanvas: MathCanvas: Intrinsic Visual Chain-of-Thought for Multimodal Mathematical Reasoning
>
> [3] MAVIS: Mathematical Visual Instruction Tuning with an Automatic Data Engine
>
> [4] GeoDANO: Geometric VLM with Domain Agnostic Vision Encoder
>
> [5] DynaMath: DynaMath: A Dynamic Visual Benchmark for Evaluating Mathematical Reasoning Robustness of Vision Language Models
>
> [6] TripletCLIP: TripletCLIP: Improving Compositional Reasoning of CLIP via Synthetic Vision-Language Negatives

---

> > ### Comment · Reviewer_xduf · 2025-11-26
> >
> > Thank you to the authors for the time spent preparing the rebuttal. However, my main concerns remain unaddressed. I agree that incorporating hard-negative contrastive learning into geometric visual encoders is beneficial. but, the current manuscript reads more like a technical report describing a domain-specific synthetic data engine, rather than a research paper that offers methodological novelty or new insights capable of inspiring further work.
> >
> > I also encourage the authors to conduct a more comprehensive review of prior literature on geometry-aware code-based diagram generation, as several relevant works have already explored this direction:
> >
> > AutoGeo [1] proposes an automatic geometry sample generation pipeline with a rich geometric clause system, specifically addressing the limitations of GPT-4–based code generation for geometric diagrams.
> >
> > We-Math [2] leverages GeoGebra, a dynamic mathematics platform that supports precise and reproducible geometric constructions through programmatic (XML-based) scripts.
> >
> > MV-MATH [3] integrates programmatic geometry diagram generation into multi-visual mathematical contexts, tackling the challenge of coordinating multiple geometric diagrams with textual reasoning.
> >
> > These works demonstrate that program-driven geometric diagram generation is not new and has been studied in several forms.
> >
> > [1] Zihan Huang, Tao Wu, Wang Lin, Shengyu Zhang, Jingyuan Chen, and Fei Wu. "AutoGeo: Automating Geometric Image Dataset Creation for Enhanced Geometry Understanding."
> >
> > [2] Runqi Qiao, Qiuna Tan, et. al. "We-Math: Does Your Large Multimodal Model Achieve Human-like Mathematical Reasoning?"
> >
> > [3]  Peijie Wang, Zhong-Zhi Li, et. al. "MV-MATH: Evaluating Multimodal Math Reasoning in Multi-Visual Contexts."

---

> > > ### Author Response · Authors · 2025-11-26
> > >
> > > Thank you for further clarifying the criterion of methodological novelty—namely, whether a contribution can support and inspire future research. Below we explain why our geometry-aware code diagrams provide practical value for future research.
> > >
> > > ## 1. Why our geometry-aware code diagram generation method matters for future multimodal reasoning
> > >
> > > A core difficulty in multimodal geometry reasoning is the high rate of **visual errors** in current LMMs (Large Multimodal Models). For example, GPT-4o exhibits a 56.8% diagram-related error rate (Figure 4 in MM-Math[1]), and Claude 3.5 Sonnet shows a 42% visual-perception error (Figure 5 in MV-MATH[2]). Thus, improving only textual reasoning from the LMMs’ output is insufficient. LMMs need reliable visual signals to improve problem-solving performance.
> > >
> > > Next-generation multimodal models are moving toward **text–image unified generation**, such as Bagel[3] or Hunyuan 3.0[4]. For geometry problem-solving, this means a model simultaneously performs mathematical reasoning **while generating editable and accurate geometric diagrams** (e.g., a diagram with “Extend CF to point D”). These scenarios require diagrams that (1) **match the real problem and solutions**, (2) are **editable**, and (3) are of **high accuracy**. Rule-based geometric primitives methods, such as AutoGeo[5], GeoDANO[6], and MAVIS[7], alone cannot satisfy these requirements.
> > >
> > > Therefore, three capabilities are essential for unified models, and our geometry-aware code diagrams provide all of them:
> > >
> > > 1.  **Construct diagrams directly from real geometry problems and their data**—otherwise the diagrams cannot be used for mathematical reasoning.
> > > 2.  **Full editability**—manual annotation cannot scale.
> > > 3.  **Accurate diagram construction**—to facilitate text-diagram alignment.
> > >
> > >
> > >
> > > ## 2. Comparing with prior  related works
> > >
> > > **AutoGeo** [5] constructs geometric diagrams **away from mathematical reasoning**: it randomly samples geometric elements to create diagrams, similar to MAVIS[7]. This leads to non-reasoning-aligned diagram and considerable noise. For example, in its Figure 1[5], the “hard text” is incorrect: the diagram does not show $\angle BAE = \angle ECB$, and point G is absent. In contrast, our Figure 2 (caption / positive pairs) effectively suppresses noise and achieves high diagram accuracy.
> > >
> > > **We-Math.**[9]
> > > Are you referring to **We-Math 2.0**[8]? The original We-Math[9] does not use GeoGebra (its diagrams are collected, not generated).  Although We-Math 2.0 is concurrent with our work, our method still precedes it. Methodologically, its problem ,solution and diagrams all require manual verification, resulting in a **low automation**. In contrast, our pipeline starts from real-world problems and solutions and automatically constructs diagrams that closely resemble the original ones, as shown in Figure 6 in the appendix of our paper.
> > >
> > > **MV-MATH** does not perform diagram generation. It only uses the Mathpix API for filtering images and does not include any programmatic generation pipeline.
> > >
> > > We hope the contributions of our proposed method to future work, together with its comparison to prior works, can demonstrate the novelty of our work.
> > >
> > >
> > >
> > >
> > > [1] MM-MATH: Advancing Multimodal Math Evaluation with Process Evaluation and Fine-grained Classification
> > >
> > > [2]MV-MATH: Evaluating Multimodal Math Reasoning in Multi-Visual Contexts
> > >
> > > [3] Emerging Properties in Unified Multimodal Pretraining
> > >
> > > [4] HunyuanImage 3.0 Technical Report
> > >
> > > [5] AutoGeo: Automating Geometric Image Dataset Creation for Enhanced Geometry Understanding
> > >
> > > [6] GeoDANO: Geometric VLM with Domain Agnostic Vision Encoder
> > >
> > > [7] MAVIS: Mathematical Visual Instruction Tuning with an Automatic Data Engine
> > >
> > > [8] WE-MATH 2.0: A Versatile MathBook System for Incentivizing Visual Mathematical Reasoning
> > >
> > > [9] We-Math: Does Your Large Multimodal Model Achieve Human-like Mathematical Reasoning?

---

> ### Author Response · Authors · 2025-11-17
>
> ## Q2: Comparing with prior works
>
> Thank you for listing these works. Below, we clarify the differences from both methodological and experimental perspectives.
> ### Prior works focus on primitive-level or formal geometric language, whereas our method directly covers real exam-style diagrams.
> GEOX[1] models diagrams using a formal language describing points, lines, and basic constraints.
> GeoDANO[2] and Primitive Vision[3] also rely on primitive-level geometric elements to construct diagrams.
> Such representations differ greatly from real exam-style geometric diagrams—they typically cannot capture full geometric constraints such as analytic-geometry reasoning or triangle-similarity reasoning.
> In contrast, our pipeline reconstructs diagrams directly from authentic exam questions and solutions, enabling coverage of a much broader set of geometric constraints beyond primitive-level modeling.
> ### Experimental comparison: Code perturbation vs. rule-designed negatives
> As discussed in Sec. 4.2.2 (“Code Perturbation vs. Rule-design”) and Appendix D, rule-designed negatives (based on primitive-level geometric elements) often fail to respect the global constraints of exam-style diagrams, which limits their effectiveness as hard negatives.
> Our code-perturbation images, derived from real-exam geometric questions, better preserve the full geometric constraints of the original problems.
> ### Our work does not design or rely on object-level detectors
>
> We do not rely on any object-level detectors; therefore, methods such as Grounding DINO or Faster R-CNN are not comparable to our setting.
>
> [1] GeoX: Geometric Problem Solving Through Unified Formalized Vision-Language Pre-training
>
> [2] GeoDANO: Geometric VLM with Domain Agnostic Vision Encoder
>
> [3] Primitive Vision: Improving Diagram Understanding in MLLMs

---

> ### Author Response · Authors · 2025-11-24
>
> ## Q3  Evaluation on Larger-Scale and More Complex Benchmarks (DynaMath)
> Thank you for raising this concern.
>
> DynaMath  is a mathematical reasoning benchmark that includes geometric element diagrams, we additionally evaluate our models on this benchmark. All experiments use **MMGeoLM-Qwen2.5-7B**, and ablation studies vary only the vision encoder. The evaluations marked with an ”*“ are taken from the original papers for performance comparison.
>
> | Model                                         | Total    | Statistics | Analytic Geometry | Arithmetic | Graph Theory | Plane Geometry | Algebra  | Scientific Figure | Solid Geometry | Puzzle Test |
> | --------------------------------------------- | -------- | ---------- | ----------------- | ---------- | ------------ | -------------- | -------- | ----------------- | -------------- | ----------- |
> | GPT-4o*                                       | 63.7     | 69.3       | 61.0              | 61.5       | 58.1         | 56.8           | 76.9     | 62.4              | 52.0           | 51.8        |
> | Qwen2.5-VL                                    | 40.6     | 35.5       | 38.7              | 10.0       | 66.8         | 45.3           | 21.8     | 45.5              | 60.3           | 25.0        |
> | Random negative                               | 39.7     | 31.0       | 45.7              | 15.0       | 60.3         | 53.8           | 22.7     | 36.6              | 25.5           | 25.0        |
> | Image-based negative                          | 48.6     | 40.3       | 53.2              | 20.0       | 72.1         | 68.2           | 30.3     | 50.3              | 30.1           | 25.0        |
> | Text-based negative (rule + retrieval top-10) | 50.1     | 53.2       | 49.3              | 15.0       | 69.2         | 65.4           | 33.3     | 51.3              | 32.3           | 25.0        |
> | All negatives                                 | 59.6 | 64.4   | 59.3          | 20.0       | **65.7**         | **70.1**       | 35.6 | 62.4          | 35.6       | 25.0        |
>
> Image-based and text-based hard negatives consistently improve performance over Qwen2.5-VL.
> The All negatives configuration achieves the close results with GPT-4o on DynaMath.
>
> Performance in categories such as **plane geometry** and **graph theory** remains high, as these tasks directly benefit from improved **geometric element understanding**.
> By contrast, categories like **arithmetic** show limited improvement, which is expected because they are unrelated to geometric perception.
>
> While our goal is not to outperform very large proprietary models, the improvements observed on DynaMath demonstrate that our hard-negative training method remains effective under more complex reasoning settings. This helps clarify the applicability of our approach beyond the original benchmarks.

---

### Official Review · Reviewer_47ne · 2025-11-01

**Soundness:** 3
**Presentation:** 3
**Contribution:** 2
**Rating:** 4
**Confidence:** 4

**Summary:**

This paper tackles the poor performance of Large Multimodal Models (LMMs) on fine-grained geometric reasoning tasks. The authors argue that standard contrastive learning fails because it relies on general descriptions, lacking geometric precision.

The authors introduce a hard negative contrastive learning framework to train the vision encoder (CLIP). This framework generates high-difficulty negative samples:
- Image-based Hard Negatives: They generate diagram code and then perturb this code to create images that are visually similar to the original but geometrically incorrect (e.g., wrong angles, broken parallelism).
- Text-based Hard Negatives: They use rule-based methods to alter geometric descriptions to be incorrect.

Experiments on three geometric reasoning benchmarks show that MMGeoLM outperforms other open-source models and reportedly rivals closed-source models like GPT-4o.

**Strengths:**

- The paper introduces a practical method for improving geometric reasoning in MLMMs. The core idea of using perturbed diagram generation code to create high-quality image-based hard negatives is technically sound.

- The proposed approach (MMGeoLM) demonstrates significant performance gains, reportedly outperforming similarly sized open-source models on three relevant geometric reasoning benchmarks.

**Weaknesses:**

- The core novelty is a clever data generation/curation technique. The method of perturbing code to create image negatives and using rules for text negatives is a form of data augmentation. While effective, the contribution is fundamentally a sophisticated form of data augmentation or data engineering, which limit its conceptual novelty and contribution.
- The empirical evaluation could be significantly strengthened. The chosen baselines are not fully representative of the current state-of-the-art; many comparisons are against models from 2024 or early 2025. Key contemporaneous works (e.g., MiMo-VL https://arxiv.org/abs/2506.03569) are notably absent, making it difficult to accurately assess the method's relative advantage. Furthermore, the evaluation would be more comprehensive if it included other challenging, relevant benchmarks, such as MathVision (https://huggingface.co/datasets/MathLLMs/MathVision), to fully validate the model's geometric reasoning capabilities.

**Questions:**

- To accurately contextualize the paper's strong results, I suggest that the authors provide a comparison against more recent and powerful open-source LMMs, such as MimoVL and any other relevant SOTA models from 2025. Besides, I encourage the authors to evaluate MMGeoLM on the MathVision benchmark. This is a well-regarded and challenging benchmark for fine-grained visual-mathematical reasoning that would provide a more comprehensive validation of the proposed method.
- What is the performance if using a "simpler" hard negative (e.g., random visual perturbations, or negatives from a standard CLIP retrieval) compared to the proposed code-perturbation method? How critical is the combination of both image-based and text-based hard negatives? What is the performance with only one of them?

---

> ### Author Response · Authors · 2025-11-19
>
> ## Q1 – On the concern that data augmentation or data engineering technique limits the contribution
> Thank you for raising this concern. Our overall framework is an LMM system, and the data augmentation or data engineering components primarily serve to match the contrastive learning paradigm. Beyond the contributions stated in the paper, we explain additional value provided by our data augmentation technique.
>
> The core contribution of our work lies in **automatically constructing high-quality, geometry-aware code diagrams**. These diagrams are not only used to generate image-based hard negatives in this work, but can also be directly applied to recent unified multimodal models that **generate geometric diagrams**. Recent “reason-while-drawing” works such as Zebra[1] and MathCanvas[2] also highlight the need for automatically constructed geometry-aware diagram. In contrast, our method produces code diagrams that can be further edited—e.g., generating auxiliary line diagrams. We will release these code-based geometric diagram datasets in the next steps.
> Below we summarize the characteristics and extended applicability of our method.
> ### A fully automated pipeline that constructs high-quality, geometry-faithful diagrams directly from real exam problems without human annotation
> Our pipeline starts from real exam questions, producing diagrams that closely match **authentic geometric problem**, as shown in Figure 4 of the main paper and Appendix Figure 6.
>
> Benchmarks such as DynaMath [3] rely on human annotators to write, execute, and validate diagram-generating code.
> In contrast, our method uses LLMs to parse problems, generate executable Python code and auto-correct syntax without any human annotation, while still achieving high geometric fidelity.
>
> ### Recent unified multimodal models require high-quality geometry diagrams, and our method supports code-based editing
>
> Unified multimodal models, when solving math problems, generate geometric diagrams which must remain undistorted and must support diagram-based reasoning (e.g., constructing auxiliary lines, folding, rotation) and text alignment (e.g., matching reasoning steps such as “AC = 4 cm”). These high-quality diagrams are essential for reason-while-drawing frameworks—such as Zebra [1] and MathCanvas [2]—which depend on accurate and diverse geometric diagrams. Our **automatically generated code-based diagrams** can be directly edited to support this capability.
>
> [1] Zebra: Zebra-CoT: A Dataset for Interleaved Vision Language Reasoning
>
> [2] MathCanvas: MathCanvas: Intrinsic Visual Chain-of-Thought for Multimodal Mathematical Reasoning
>
>
> [3] DynaMath: DynaMath: A Dynamic Visual Benchmark for Evaluating Mathematical Reasoning Robustness of Vision Language Models

---

> ### Author Response · Authors · 2025-11-24
>
> ## Q2 Experiments with SOTA models and evaluations on other benchmarks
> The evaluations marked with an ”*“ are taken from the original papers for performance comparison.
> ### We conducted GeoQA and MM-Math evaluation experiments on MiMo-VL-7B, GPT-4o, and MMGeoLM.
>
> | Model | GeoQA | Easy | Mid | Hard | Overall |
> |-------|-------|------|-----|------|---------|
> |Human* |92.3 |90.7 |81.9| 47.6|80.4
> | GPT-4o* | 58.9 | 45.8 | 30.0 | 10.9 | 31.8 |
> | MiMo-VL-7B | 89.2 | 67.0 | 64.1 | 30.0 | 63.9 |
> | MMGeoLM-Qwen2.5-7B* | 69.2 | 55.3 | 36.9 | 9.0 | 39.2 |
>
> MiMo-VL-7B, as a reinforcement–enhanced reasoning model, reaches 89.2 on GeoQA and shows large gains over MMGeoLM on the mid and hard subsets.
> However, on easy problems, it exceeds MMGeoLM by only 11.7, still far from the human score of 90.7.
>
> To further investigate error sources and the technical contribution, we follow the MM-Math[1]'s method and extract the first error from each model’s answer, classifying it into four categories: (1) image-misunderstanding (2) reasoning error (3) calculation error and (4) misunderstanding the question
>
> | Model | Image | Reasoning | Calculation | Question |
> |-------|-------|-----------|-------------|----------|
> | GPT-4o* | 56.6 | 36.4 | 3.4 | 3.6 |
> | MiMo-VL-7B | 75.8 (+19.2%) | 12.8 (-23.6%) | 5.8 | 5.6 |
> | MMGeoLM-Qwen2.5-7B | 28.3 (-28.3%) | 57.1(+20.7) | 8.6 | 6.0 |
>
> MiMo-VL-7B significantly reduces reasoning errors, while the main issue becomes image misunderstanding, reaching 75.8%.
> In contrast, our MMGeoLM‘s **image misunderstanding drops to 28.3**, half of GPT-4o’s.
>
> our method mainly improves **the vision encoder’s ability to understand geometric elements**, making it complementary to MiMo’s reinforcement reasoning improvements.
>
> ### Performance on Other Benchmarks
>
> On MathVision, our model performs poorly: MMGeoLM-Qwen2.5-7B achieves only 5.6% ACC.
>
> Upon analysis, we find that MathVision focuses on direct object visual recognition, such as counting how many ducks are in the picture, which strongly differs from exam-style geometric element reasoning.
> Since DynaMath [2] is also a mathematical reasoning benchmark that contains geometric elements diagrams, we evaluate our models on this benchmark. All experiments use MMGeoLM-Qwen2.5-7B, while the ablation studies vary only the vision encoder.
>
>
> | Model | Total | Statistics | Analytic Geometry | Arithmetic | Graph Theory | Plane Geometry | Algebra | Scientific Figure | Solid Geometry | Puzzle Test |
> |-------|-------|------------|-------------------|------------|--------------|----------------|---------|-------------------|----------------|-------------|
> | GPT-4o* | 63.7 | 69.3 | 61.0 | 61.5 | 58.1 | 56.8 | 76.9 | 62.4 | 52.0 | 51.8 |
> | Qwen2.5-VL | 40.6 | 35.5 | 38.7 | 10.0 | 66.8 | 45.3 | 21.8 | 45.5 | 60.3 | 25.0 |
> | Random negative | 39.7 | 31.0 | 45.7 | 15.0 | 60.3 | 53.8 | 22.7 | 36.6 | 25.5 | 25.0 |
> | Image-based negative | 48.6 | 40.3 | 53.2 | 20.0 | 72.1 | 68.2 | 30.3 | 50.3 | 30.1 | 25.0 |
> |Text-based negative (rule-based + retrieval top 10) | 50.1 | 53.2 | 49.3 | 15.0 | 69.2 | 65.4 | 33.3 | 51.3 | 32.3 | 25.0 |
> | All negatives | 59.6 | 64.4 | 59.3 | 20.0 | 65.7 | 70.1 | 35.6 | 62.4 | 35.6 | 25.0 |
>
> Image-based and text-based hard negatives consistently improve reasoning performance compared to Qwen2.5-VL.
> The All negatives setting yields the largest overall gain, closing to GPT-4o on DynaMath.
>
> Performance on certain categories—such as plane geometry and graph theory—remains high. This is mainly because these categories benefit directly from **enhanced geometric element understanding** provided by our method.
> In contrast, for arithmetic, our method does not provide effective improvements, as this category is unrelated to geometric perception.
>
> [1] MM-MATH: Advancing Multimodal Math Evaluation with Process Evaluation and Fine-grained Classification
>
> [2] DynaMath: A Dynamic Visual Benchmark for Evaluating Mathematical Reasoning Robustness of Vision Language Models

---

> ### Author Response · Authors · 2025-11-24
>
> ## Q3 – Clarification on simpler hard negatives and the combination of image/text negatives
> Thank you for raising this concern. We provide further clarification below.
> ### Random visual perturbations vs. our code-perturbation method
>
> | In-Batch Negative | GeoQA | MM-MATH Easy | MM-MATH Med | MM-MATH Hard | MM-MATH Avg |
> |-------------------|-------|--------------|-------------|--------------|-------------|
> | Random | 46.2 | 41.8 | 23.7 | 4.5 | 25.5 |
> | Image-based NS | 57.3 | 51.7 | 29.0 | 4.5 | 32.1 |
>
> As shown in Table 6 (“Hard Negative Performance Gains or Trade-off for In-Batch Training”), our code-perturbation method consistently improves performance on GeoQA and on the easy, medium, and overall categories of MM-Math.
> The main reason is that code-perturbation naturally produces strong mutual hard negatives within a batch.
>
> For each image-based negative, our method also generates a corresponding caption, and the perturbed diagrams contain subtle but meaningful geometric differences (see Appendix Figure 6).
> When these hard negatives are placed in the same batch, they serve as mutual hard negatives, yielding larger gains than random visual perturbations.
> ### How critical is combining image-based and text-based negatives? What if we use only one of them?
>
> We also conducted ablation experiments, reported in Section 4.2.1 (Table 3).
>
> | Vision Encoder | #Num. | GeoQA | MM-MATH Easy | MM-MATH Med | MM-MATH Hard | MM-MATH Avg |
> |----------------|-------|-------|--------------|-------------|--------------|-------------|
> | Original | - | 45.4 | 40.5 | 20.8 | 4.5 | 23.8 |
> | Random10 | 400K | 45.0 | 39.8 | 22.7 | 4.5 | 24.9 |
> | Retrieval10 | 100K | 53.0 | 43.6 | 24.8 | 4.5 | 26.6 |
> | Rule10 | 100K | 48.8 | 45.4 | 27.6 | 4.5 | 28.1 |
> | Image-based10 | 4K | 54.9 | 45.6 | 26.5 | 4.5 | 29.0 |
> | Retrieval10+Rule10 | 200K | 56.5 | 47.5 | 28.3 | 4.5 | 29.4 |
> | All Negatives | 204K | 58.2 | 49.8 | 29.1 | 4.5 | 30.4 |
>
> Using either text-based negatives or image-based negatives alone improves performance compared to the standard CLIP baseline. (Retrieval, Rule, Image-base vs Random)
>
> Combining both image-based and text-based negatives leads to further performance gains, demonstrating that the two types of hard negatives are complementary rather than interfering with each other.

---

### Official Review · Reviewer_GgP3 · 2025-11-01

**Soundness:** 2
**Presentation:** 3
**Contribution:** 3
**Rating:** 6
**Confidence:** 4

**Summary:**

The paper introduces MMGeoLM, a Large Multimodal Model (LMM) designed to enhance fine-grained geometric understanding through a novel hard negative contrastive learning framework. This framework integrates image-based contrastive learning using generation-based hard negatives and text-based contrastive learning employing rule-based and retrieval-based negatives. The authors train a vision encoder (CLIP) with this method, resulting in MMGeoLM, which demonstrates superior performance on three geometric reasoning benchmarks, even at a 7B parameter scale, rivaling models like GPT-4o. Ablation studies provide insights into optimizing the training pipeline for geometric reasoning tasks.

**Strengths:**

1. The paper presents a novel hard negative contrastive learning framework that effectively addresses the challenge of capturing fine-grained visual differences in geometric scenarios. By combining image-based and text-based contrastive learning, the approach enhances the model's ability to understand complex geometric relationships.
2. The integration of generation-based hard negatives and rule-based and retrieval-based text negatives is a unique contribution.

**Weaknesses:**

1. While the model performs well on the selected benchmarks, the paper does not discuss its performance on a broader range of geometric reasoning tasks. It would be beneficial to understand how MMGeoLM generalizes to other problem domains.
2. The paper lacks comparisons with other state-of-the-art models in the field. Including such comparisons would provide a clearer context for evaluating MMGeoLM's performance and highlight its relative advantages.

**Questions:**

1. Can you provide more details on the generation-based hard negatives and how they are created?
2. How does MMGeoLM perform on geometric reasoning tasks beyond the three benchmarks presented?
3. What are the computational requirements for training MMGeoLM, and how does it scale with larger datasets?

---

> ### Author Response · Authors · 2025-11-24
>
> ## Q1&Q2 Experiments with SOTA models and evaluations on other benchmarks
>
> ### We conducted GeoQA and MM-Math evaluation experiments on MiMo-VL-7B, GPT-4o, and MMGeoLM.
>
> Thank you for your concern and suggestion. We include comparisons with the SOTA model MiMo-VL-7B, a reinforcement-enhanced reasoning model, and GPT-4o. The evaluations marked with an ”*“ are taken from the original papers for performance comparison.
>
>
>
> | Model | GeoQA | Easy | Mid | Hard | Overall |
> |-------|-------|------|-----|------|---------|
> |Human* |92.3 |90.7 |81.9| 47.6|80.4
> | GPT-4o* | 58.9 | 45.8 | 30.0 | 10.9 | 31.8 |
> | MiMo-VL-7B | 89.2 | 67.0 | 64.1 | 30.0 | 63.9 |
> | MMGeoLM-Qwen2.5-7B* | 69.2 | 55.3 | 36.9 | 9.0 | 39.2 |
>
> MiMo-VL-7B, as a reinforcement–enhanced reasoning model, reaches 89.2 on GeoQA and shows large gains over MMGeoLM on the mid and hard subsets.
> However, on easy problems, it exceeds MMGeoLM by only 11.7, still far from the human score of 90.7.
>
> To further investigate error sources and the technical contribution, we follow the MM-Math[1]'s method and extract the first error from each model’s answer, classifying it into four categories: (1) image-misunderstanding (2) reasoning error (3) calculation error and (4) misunderstanding the question
>
> | Model | Image | Reasoning | Calculation | Question |
> |-------|-------|-----------|-------------|----------|
> | GPT-4o* | 56.6 | 36.4 | 3.4 | 3.6 |
> | MiMo-VL-7B | 75.8 (+19.2%) | 12.8 (-23.6%) | 5.8 | 5.6 |
> | MMGeoLM-Qwen2.5-7B | 28.3 (-28.3%) | 57.1(+20.7) | 8.6 | 6.0 |
>
> MiMo-VL-7B significantly reduces reasoning errors, while the main issue becomes image misunderstanding, reaching 75.8%.
> In contrast, our MMGeoLM‘s **image misunderstanding drops to 28.3**, half of GPT-4o’s.
>
> our method mainly improves **the vision encoder’s ability to understand geometric elements**, making it complementary to MiMo’s reinforcement reasoning improvements.
>
> ### Performance on Other Benchmarks
>
>
> DynaMath [2] is also a mathematical reasoning benchmark that contains geometric elements diagrams, we evaluate our models on this benchmark. All experiments use MMGeoLM-Qwen2.5-7B, while the ablation studies vary only the vision encoder.
>
>
> | Model | Total | Statistics | Analytic Geometry | Arithmetic | Graph Theory | Plane Geometry | Algebra | Scientific Figure | Solid Geometry | Puzzle Test |
> |-------|-------|------------|-------------------|------------|--------------|----------------|---------|-------------------|----------------|-------------|
> | GPT-4o* | 63.7 | 69.3 | 61.0 | 61.5 | 58.1 | 56.8 | 76.9 | 62.4 | 52.0 | 51.8 |
> | Qwen2.5-VL | 40.6 | 35.5 | 38.7 | 10.0 | 66.8 | 45.3 | 21.8 | 45.5 | 60.3 | 25.0 |
> | Random negative | 39.7 | 31.0 | 45.7 | 15.0 | 60.3 | 53.8 | 22.7 | 36.6 | 25.5 | 25.0 |
> | Image-based negative | 48.6 | 40.3 | 53.2 | 20.0 | 72.1 | 68.2 | 30.3 | 50.3 | 30.1 | 25.0 |
> |Text-based negative (rule-based + retrieval top 10) | 50.1 | 53.2 | 49.3 | 15.0 | 69.2 | 65.4 | 33.3 | 51.3 | 32.3 | 25.0 |
> | All negatives | 59.6 | 64.4 | 59.3 | 20.0 | 65.7 | 70.1 | 35.6 | 62.4 | 35.6 | 25.0 |
>
> Image-based and text-based hard negatives consistently improve reasoning performance compared to Qwen2.5-VL.
> The All negatives setting yields the largest overall gain, closing to GPT-4o on DynaMath.
>
> Performance on certain categories—such as plane geometry and graph theory—remains high. This is mainly because these categories benefit directly from **enhanced geometric element understanding** provided by our method.
> In contrast, for arithmetic, our method does not provide effective improvements, as this category is unrelated to geometric perception.
>
> [1] MM-MATH: Advancing Multimodal Math Evaluation with Process Evaluation and Fine-grained Classification
>
> [2] DynaMath: A Dynamic Visual Benchmark for Evaluating Mathematical Reasoning Robustness of Vision Language Models

---

> ### Author Response · Authors · 2025-11-24
>
> ## Q3 Generation-Based Hard Negatives
> Our image-based hard negatives are produced through a multi-step process that begins with real exam problems and yields geometric positive and hard negative image.
>
> 1.	**Data collection**
> We collect middle-school and high-school math problems containing diagrams, covering analytic geometry, solid geometry, and plane geometry.
>
> 2.	**Code generation from problem text.**
> We feed the problem statement and its solution into Gemini 2.5 Pro using the prompt in Appendix Figure 10.
> The model outputs Python code for drawing the diagram.
> We execute the code and use GLM4.5 to automatically fix errors (due to lower cost), as shown in Appendix Figure 14, producing the positive image.
>
> 3.	**Positive caption generation.**
> We feed the Python code to Gemini 2.5 Pro to produce a positive caption using the prompt in Appendix Figure 11,.
> This caption describes only the geometric elements in the code and does not require reasoning.
>
> 4.	**Negative caption generation.**
> We modify the positive caption into a negative caption using the prompt in Appendix Figure 12 and GLM-4.5.
>
> 5.	**Negative image generation**
> Based on the negative caption, we  modify the positive Python code to obtain the negative image, forming a geometry-faithful hard negative pair.

---

> ### Author Response · Authors · 2025-11-24
>
> ## Q4 Compute requirements and scalability
> MMGeoLM is trained in three stages, each with moderate computational requirements:
>
> **MMCLIP training (vision encoder)**: 0.5B AltCLIP backbone; trained on 59M aligned pairs + hard negatives.
> Training takes on 1 A100 GPUs.
>
> **MLP adapter training**: only a 2-layer MLP (≈ 15M parameters); negligible compute.
>
> **SFT on the LLM**: 7B LLM (MAmmoTH2 / Qwen2.5); trained on 17K and 300K math instructions. Training takes on 4 A100 GPUs.
>
> Because each stage trains only a subset of the model (vision encoder or adapter—not the whole LLM), the overall compute cost is significantly lower than full LLM finetuning.
>
> ## Scaling with larger datasets
> The table below shows how MM-MATH accuracy improves when scaling:
>
> •	SFT data from 7K → 17K → 434K
>
> •	Vision encoder hard negatives from random → retrieval/rule → retrieval+rule+image
>
> •	Base LLM from Vicuna → Mammoth2 → Qwen2.5
>
> | Clip Training | MLP Training | SFT Data | Base LLM | MM-MATH Score |
> |---------------|-------------|----------|----------|---------------|
> | Mavis Align 590K | Mavis Align 590K | 7K | Vicuna | 11.7% |
> | Mavis Align 590K | Mavis Align 590K | 17K + MAVIS | Vicuna | 17.8% |
> | Mavis Align 590K | Mavis Align 590K | 17K + MAVIS | Mammoth2 | 24.2% |
> | Mavis Align 590K | GLLaVA 67K | 17K + MAVIS | Mammoth2 | 26.8% |
> | Mavis Align 590K + Retrieval + Rule | GLLaVA 67K | 17K + MAVIS | Mammoth2 | 32.4% |
> | Mavis Align 590K + Retrieval + Rule + Image | GLLaVA 67K | 17K + MAVIS | Mammoth2 | 36.9% |
> | Mavis Align 590K + Retrieval + Rule + Image | GLLaVA 67K | 17K + MAVIS | QWen2.5 | 39.2% |
>
> These results indicate that MMGeoLM benefits from training data and stronger base LLMs, while remaining computationally reasonable.
>
> Regarding further dataset scaling, extending beyond 17K high-quality SFT examples is difficult in practice: each sample costs approximately 2 RMB to produce, and collecting hundreds of thousands of such samples exceeds our economic capacity.

---

### Meta-Review · Area_Chair_pXeb · 2026-01-08

**Summary:**

This paper focuses on the capability of large multimodal models to understand fine-grained geometric information. The core idea is to automatically generate hard negatives from both the image and text modalities for contrastive learning, and therefore getting better representations for improving large multimodal models. Experimental results show promising improvements over the baselines.

AC agrees with the following concerns:
- The contribution is primarily about data engineering for a specific type of geometric task, which is somewhat limited. The paper could be strengthened by expanding its scope, for example by modifying or extending the proposed data augmentation method to broader geometric understanding and reasoning tasks, or even beyond.
- Including complete results on broader benchmarks for comprehensive comparison, such as MathVision and the full version of MathVista

**Reviewer Concerns:**

- Limited novelty and scope.
- (Partially addressed) Including results of more advanced baselines.
  - New results are provided, but not for all datasets.
- (Partially addressed) Including results of more benchmarks.
  - Results on one additional benchmark are reported. The benchmark mentioned by the reviewers seens to have poor performance.
- (Fully addressed) Additional experiments comparing the proposed hard negatives with simply generated hard negatives.
- (Fully addressed) Evaluation of whether hallucination is reduced.
- (Fully addressed) Comparison with additional prior work.

**Reviewer Scores:**

Based on my evaluation, I believe Reviewer xduf and Reviewer q6Rm might have the chance to increase their scores. Reviewer 47ne and Reviewer GgP3 are less likely to change the scores. Overall rating for this paper might be still negative.

---

### Decision · Program_Chairs · 2026-01-26

Reject